

**Optical properties and molecular compositions of water-soluble and water-**
**insoluble brown carbon (BrC) aerosols in Northwest China**
Jianjun Li[1,2], Qi Zhang[2,*], Gehui Wang[1,3,4,*], Jin Li[1], Can Wu[1,3], Lang Liu[1], Jiayuan Wang[1,2],
Wenqing Jiang[2], Lijuan Li[1,2], Kin Fai Ho[1,5], Junji Cao[1]
[1] Key Lab of Aerosol Chemistry & Physics, SKLLQG, Institute of Earth Environment, Chinese
Academy of Sciences, Xi'an 710061, China
[2] Department of Environmental Toxicology, University of California, Davis, CA 95616, USA
[3] Key Laboratory of Geographic Information Science of the Ministry of Education, School of
Geographic Sciences, East China Normal University, Shanghai 200241, China
[4] Institute of Eco-Chongming, 3663 N. Zhongshan Rd., Shanghai 200062, China
[5] The Jockey Club School of Public Health and Primary Care, The Chinese University of Hong
Kong, Hong Kong, China
*Corresponding authors:
Prof. Qi Zhang
Department of Environmental Toxicology, University of California, Davis
One Shields Avenue, Davis, CA 95616
Phone: 1-530-752-5779
Fax: 1-530-752-3394
Email: dkwzhang@ucdavis.edu;
Prof. Gehui Wang
School of Geographic Sciences, East China Normal University, Shanghai, China
500 Dongchuan Rd., Shanghai 200241, China
Phone: 86-21-5434-1193
E-mail: ghwang@geo.ecnu.edu.cn.





**Abstract**
Brown carbon (BrC) contributes significantly to aerosol light absorption, thus can affect the earth's
radiation balance and atmospheric photochemical processes. In this study, we examined the light
absorption properties and molecular compositions of water-soluble (WS) and water-insoluble (WI)
BrC in $PM_{2.5}$ collected from a rural site in the Guanzhong Basin – a highly polluted region in
Northwest China. Both WS-BrC and WI-BrC showed elevated light absorption coefficients (Abs)
in winter (4-7 times of those in summer) mainly attributed to enhanced emissions from residential
biomass burning (BB) for heating. While the average mass absorption coefficients at 365 nm
($MAC_{365}$) of WS-BrC were similar between daytime and nighttime in summer (0.99±0.17 and
1.01±0.18 $m^2\ g^{-1}$, respectively), the average $MAC_{365}$ of WI-BrC was more than a factor of 2 higher
during daytime (2.45±1.14 $m^2\ g^{-1}$) than at night (1.18±0.36 $m^2\ g^{-1}$). This difference was mainly
attributed to enhanced photochemical formation of WI-BrC species, such as oxygenated polycyclic
aromatic hydrocarbons (OPAHs). In contrast, the MACs of WS-BrC and WI-BrC were generally
similar in winter and both showed little diel differences. The Abs of WS-BrC correlated strongly
with relative humidity, sulfate, and $NO_2$, suggesting that aqueous-phase reactions is an important
pathway for secondary BrC formation during the winter season in Northwest China. Nitrophenols
on average contributed 2.44±1.78% of the Abs of WS-BrC in winter, but only 0.12±0.03% in
summer due to faster photodegradation reactions. WS-BrC and WI-BrC were estimated to account
for 0.83±0.23% and 0.53±0.33%, respectively, of the total down-welling solar radiation in the UV
range in summer, and 1.67±0.72% and 2.07±1.24%, respectively, in winter. The total absorption
by BrC in the UV region was about 55-79% relative to the elemental carbon (EC) absorption.
Keywords: Brown Carbon (BrC); Organic Aerosol; Optical Property; Molecular Composition





## 1. Introduction


Light-absorbing organic matter, termed as "brown carbon (BrC)", has been recognized as an
important climate forcer due to its ability to directly interact with both incoming solar radiation
and outgoing terrestrial radiation (Andreae and Gelencser, 2006;Laskin et al., 2015). BrC is a
complex mixture of organic compounds, which collectively show a light absorption profile
increasing exponentially from the visible (Vis) to the ultraviolet (UV) range. Due to the high
abundance of organic aerosol in continental regions, especially in places with intensive
anthropogenic pollution, the contribution of BrC to aerosol absorption in the near-UV range is
potentially significant (Kirillova et al., 2014b;Huang et al., 2018;Yan et al., 2015). For example, a
model study showed that BrC contributes up to +0.25 W m$^{-2}$ of radiative forcing on a planetary
scale, which is approximately 19% of the absorption by anthropogenic aerosols (Feng et al., 2013).
Moreover, the strong absorption of BrC in the UV spectral region can reduce the solar actinic flux,
and subsequently affect atmospheric photochemistry and tropospheric ozone production (Jacobson,
1998;Mohr et al., 2013).
A thorough understanding of the sources and transformation processes of BrC in the
atmosphere is important, but it is still lacking. Biomass/biofuel combustion, including forest fires,
and burning of wood and agricultural wastes for residential cooking and heating, has been shown
as a particularly important source of BrC (Washenfelder et al., 2015;Desyaterik et al., 2013;Lin et
al., 2017). BrC can also be emitted directly from coal burning (Yan et al., 2017), and biogenic
release of fungi, plant debris, and humic matter (Rizzo et al., 2013;Rizzo et al., 2011). In addition,
recent studies suggested that secondary BrC can be formed through various reaction pathways,
including photooxidation of aromatic volatile organic compounds (VOCs) (Lin et al., 2015;Liu et



al., 2016), reactive uptake of isoprene epoxydiols onto preexisting sulfate aerosols (Lin et al.,
2014), aqueous oxidation of phenolic compounds and α-dicarbonyls (Chang and Thompson,
2010;Nozière and Esteve, 2005;Smith et al., 2016;Yu et al., 2014;Xu et al., 2018), and reactions
of ammonia or amines with carbonyl compounds in particles or cloud droplets (Nozière et al.,
2007;Laskin et al., 2010;Updyke et al., 2012;Nguyen et al., 2012;De Haan et al., 2018;Powelson
et al., 2014). However, atmospheric oxidation processes may also cause "photobleach" –
photodegradation of BrC into less light-absorbing compounds (Lee et al., 2014;Romonosky et al.,
2015;Sumlin et al., 2017), which may complicate the understanding of BrC in the atmosphere.
A common way to quantify the absorption properties of BrC is to measure the absorbance of
aerosol extracts over a wide wavelength range using spectrophotometers. This approach can
differentiate the interference of black carbon (BC) or mineral dust (Hecobian et al., 2010). Most
of the studies use ultrapure water to extract organic substance in the aerosol, and thus measure the
optical properties of water-soluble BrC (WS-BrC) (Wu et al., 2019;Hecobian et al., 2010;Kirillova
et al., 2014b). In addition, some studies analyzed the light absorption of BrC extracted using polar
organic solvents such as methanol or acetone (Liu et al., 2013;Huang et al., 2018;Kim et al., 2016).
Since such extracts contain both water-soluble and water-insoluble chromophores, little
information is available regarding the contribution and formation of water-insoluble BrC (WI-
BrC). However, it is important to understand WI-BrC given the facts that some water-insoluble
organic compounds, such as polycyclic aromatic hydrocarbons and their derivatives, are effective
light absorbers and that the mass absorption of WI-BrC could be even greater than that of the
water-soluble fraction (Chen and Bond, 2010;Huang et al., 2018;Sengupta et al., 2018). Thus, it is
necessary to extract water-soluble and water-insoluble organic components separately, e.g., via



using solvents with different polarity in sequence. Combining with measurements of BrC
molecular compositions, the UV-vis absorption properties of the water-soluble and water-insoluble
extracts may help us better understand the sources and formation mechanisms of light-absorbing
compounds in the atmosphere.
China has been experiencing serious atmospheric pollution conditions in recent decades, and
both model and field results showed elevated light absorption of BrC in most regions of China
(Huang et al., 2018;Cheng et al., 2011;Yan et al., 2017;Li et al., 2016b) compared to developed
countries such as the U.S. (Hecobian et al., 2010;Washenfelder et al., 2015) and European
countries (Mohr et al., 2013;Teich et al., 2017). However, BrC-related data are scarce in the
Guanzhong Basin (Shen et al., 2017;Huang et al., 2018), which is one of the most polluted regions
in China (van Donkelaar et al., 2010). Here we present measurements of the optical properties of
WS-BrC and WI-BrC in $PM_{2.5}$ collected from a rural area of the Guanzhong Basin during winter
and summer seasons. We also measured the concentrations of several BrC compounds as well as
those of organic carbon (OC), elemental carbon (EC), water-soluble OC (WSOC) and inorganic
ions. These data were analyzed to examine the effects of sources emissions, daytime
photochemical oxidation, and aqueous-phase chemistry on WS- and WI-BrC components in
different seasons.
**2. Experimental section**
**2.1 Sample collection**
The sampling was conducted at a small village (namely Lincun, 34º44′ N and 109º32′E, 354m
a.s.l.) ~ 40 km northeast to Xi'an, the capital of Shaanxi Province (Figure S1). The sampling site
is located in the central part of Guanzhong Basin with no obvious point source of air pollutants in



the surrounding areas. PM$_{2.5}$ samples were collected twice a day (~8 am to 20 pm and ~20 pm to
8 am) onto prebaked (450 $^{\circ}$C, 6-8 hr) quartz fiber filters (Whatman, QM-A, USA) during Aug. 3-
23, 2016 and Jan. 20-Feb. 1, 2017 using a TISCH Environmental (USA) PM$_{2.5}$ high volume (1.13
m$^3$ min$^{-1}$) sampler. Field blank samples were also collected by mounting blank filters onto the
sampler for about 15 min without pumping any air. After sampling, the sample filters were
immediately sealed in aluminum foil bags, and then stored in a freezer ($-5$ $^{\circ}$C) prior to analysis.
Meteorological conditions, and concentrations of O$_3$ and NO$_2$ during this studied period are
presented in Figure 1.

**2.2 Filter extraction and absorption spectra analysis**

For each PM$_{2.5}$ sample, a portion of the filter (~13.384 cm$^2$) was first extracted in 8 ml of

Milli-Q water (18.2 M$\Omega$) through 30 min of sonication at ~ 0$^{\circ}$C. The water extract was then filtered
via vacuum filtration with a 25mm diameter 5 μm pore hydrophobic PTFE membrane filter (Merck
Millipore Ltd, Mitex$^{TM}$ Membrane Filters, USA). Afterwards, the insoluble PM components
collected on the PTFE membrane filter and remained on the sample filter were rinsed with 2 ml
Milli-Q water, air dried, and then extracted via sonication in 8 ml pure acetonitrile (ACN)
(Honeywell Burdick & Jackson, LC/MS Grade, USA). The acetonitrile extract was filtered via a
13 mm diameter 0.45 μm pore syringe filter (PALL, Bulk Acrodisc®, PTFE Membrane Filters,
USA). The light absorption spectra of the water and the acetonitrile extracts were measured
between 190 nm to 820 nm by a diode-array spectrophotometer (Hewlett Packard 8452A, USA)
using quartz cuvettes with 1 cm length path. Field blank filters were extracted and measured in the
same manner as the samples. Data presented in this study were corrected for the field blanks (<10%
relative to field samples).



**2.3 Chemical Analysis**


OC and EC were analyzed using DRI Carbon Analyzer (Model 2001, USA). Another piece
of the filter sample (~8.6 cm$^2$) was extracted with Milli-Q water (18.2MΩ), and filtered through
a PTFE syringe filter. Then the water-extract was analyzed for water-soluble inorganic ions
($SO_4^{2-}$, $NO_3^-$, $NH_4^+$, $Cl^-$, $F^-$, $Ca^{2+}$, $K^+$, $Na^+$ and $Mg^{2+}$) using a Metrohm Ion Chromatography
(Metrohm 940, Switzerland) and WSOC using a Shimadzu TOC analyzer (TOC-L CPH, Japan)
and. Concentrations of individual molecules, including levoglucosan, parent-PAHs, Oxygenated-
PAHs (OPAHs), nitrophenols, and isoprene and α-/β-pinene derived products, were measured
using GC/EI-MS (Agilent 7890A-5975C, USA) calibrated by authentic standards. More details
on these measurements can be found in previous publications (Li et al., 2014).

**2.4 Data Interpretation**


In this study, water-insoluble OC (WIOC) was calculated by the difference between OC and
WSOC:
$$M_{WIOC} = M_{OC} - M_{WSOC} \qquad (1)$$
where $M_{WIOC}$, $M_{OC}$, and $M_{WSOC}$ correspond to the mass concentration (in μgC m$^{-3}$) of WIOC,
OC, and WSOC, respectively, in the air.
The absorption coefficient of WS-BrC (Abs$_{\lambda,WS\text{-}BrC}$, Mm$^{-1}$) or WI-BrC (Abs$_{\lambda,WI\text{-}BrC}$, Mm$^{-1}$)
at a given wavelength (λ) is determined from the UV-vis spectrum of the water extract (Hecobian
et al., 2010;Laskin et al., 2015)
$$Abs_\lambda = (A_\lambda - A_{700}) \times \frac{V_{solvent}}{V_a \times l} \times \ln(10) \times 100 \qquad (2)$$
where $A_\lambda$ is the absorbance of the water (A$_{\lambda,WS\text{-}BrC}$) or ACN (A$_{\lambda,WI\text{-}BrC}$) extract at λ, V$_{solvent}$ (ml)
is the volume of solvent (water or ACN) used to extract the filter (8 mL), and Va (m$^3$) is the air


volume passed through the filter punch. $l$ (cm) is the optical length of the quartz cuvettes used for
UV-vis measurement and ln(10) is used to convert the logbase-10 (provided by the
spectrophotometer) to natural logarithm. 100 is for unit conversion. $A_{700}$ (absorbance at the
wavelength of 700 nm) is subtracted to minimize the interference of baseline shift. The mass
absorption coefficient of WS-BrC ($MAC_{\lambda,WS\text{-}BrC}$, $m^2$ $g^{-1}$) or WI-BrC ($MAC_{\lambda,WI\text{-}BrC}$, $m^2$ $g^{-1}$) at
wavelength of $\lambda$ is calculated using eq (3)
$$MAC_\lambda = \frac{Abs_\lambda}{M} \qquad (3)$$
Note that since it is possible that not all the WI-BrC was extracted into ACN, the $Abs_{\lambda,WI\text{-}BrC}$ and
$MAC_{\lambda,WI\text{-}BrC}$ reported in this study are likely the lower bound values. Nevertheless, the
underestimation is probably insignificant since Chen and Bond (Chen and Bond, 2010) reported
that >92% of BrC was extractable by organic solvents (methanol or acetone).

The wavelength dependence for BrC absorption is fit with a power law equation:

$$Abs_\lambda = K \times \lambda^{-AAE} \qquad (4)$$
where K is a constant and AAE stands for absorption Ångström exponent. In this study, the AAE
for a given sample is calculated through the linear regression of log($Abs_\lambda$) against log $\lambda$ between
300−450 nm. This wavelength range is chosen because the fits of all the samples in this study are
better than $r^2$=0.99. Note that slightly higher AAE values (by up to 10%) are obtained using a
wider wavelength range (e.g., 300-550 nm; Figure S2).

The fraction of solar irradiance absorbed by particulate BrC at a given wavelength $\lambda$ is

estimated following the Beer−Lambert's law:
$$\frac{I_0 - I}{I_0}(\lambda) = 1 - e^{-b_{ap,\lambda,x} \times h_{ABL}} \qquad (5)$$
where x denotes WS-BrC or WI-BrC, $h_{ABL}$ is the atmospheric boundary layer height (assuming





1200 m in summer and 600 m in winter) according to the assumption that the ground
measurement results are representative of the average values in the whole atmospheric boundary
layer (ABL) (Kirchstetter et al., 2004;Kirillova et al., 2014a), and $b_{ap,\lambda,x}$ corresponds to the
absorption coefficient ($b_{ap,}$ m$^{-1}$) of WS-BrC or WI-BrC at wavelength of $\lambda$. Previous studies
showed that the light absorption coefficient of particulate BrC ($b_{ap,\lambda,BrC}$) is around 0.7−2.0 times
of that from bulk solution ($Abs_{\lambda,WS-BrC \text{ or } WI-BrC}$) (Liu et al., 2013;Sun et al., 2007). Here, a
conversion factor of 1.3 is applied based on a Mie theory calculation of aerosols in Xi'an (~ 40
km away from the sampling site) (Wu, 2018). $I_0$ denotes the incident solar radiance in the form
of either actinic flux (in quanta s$^{-1}$ cm$^{-2}$ nm$^{-1}$) or irradiance (in W m$^{-2}$ nm$^{-1}$), which were obtained
using the TUV Quick Calculator (http://cprm.acom.ucar.edu/Models/TUV/Interactive_TUV/).
($I_0$-I) denotes the direct absorption of solar actinic flux or irradiance by BrC.
**3. Results and Discussion**
**3.1 Optical absorption characteristics of WS-BrC and WI-BrC**
The average absorption spectra of WS-BrC and WI-BrC ($\lambda$ = 300-700 nm) during daytime
and nighttime in different seasons are shown in Figure 2a &b. The absorption Ångström
exponents for both WS-BrC ($AAE_{WS-BrC}$) and WI-BrC ($AAE_{WI-BrC}$) are generally higher than 5,
verifying the contribution of BrC to aerosol absorptivity in the region. The average $AAE_{WS-BrC}$
are similar between summer (5.43±0.41) and winter (5.11±0.53). Huang et al. (2014) and Shen et
al. (2017) reported comparable $AAE_{WS-BrC}$ values (5.3-5.7) with no significant seasonal change
at urban sites of Xi'an, suggesting common characteristics of BrC on a regional scale in the
Guanzhong Basin of China. Comparable AAE values were reported for WS-BrC in Switzerland
(3.8-5.1) (Moschos et al., 2018) and Nepal (4.2-5.6) (Wu et al., 2019;Kirillova et al., 2016), but





higher $AAE_{WS\text{-}BrC}$ were found in Southeastern US ($7 \pm 1$) (Hecobian et al., 2010), Los Angeles
Basin ($7.6 \pm 0.5$) (Zhang et al., 2013), Korea (5.84-9.17) (Kim et al., 2016), and Beijing (7.0-7.5)
(Cheng et al., 2011).

The $AAE_{WI\text{-}BrC}$ shows more obvious seasonal variations with a higher average value in

winter ($6.04\pm0.22$) than in summer ($5.01\pm0.58$). This difference suggests that the chemical
composition of WI-BrC might be more different in different seasons, due to variations in the
sources and atmospheric formation and aging processes of light absorbing hydrophobic
compounds.

The light absorption properties of WS-BrC and WI-BrC present obvious seasonal variations

(Figure 2). The average ($\pm1\sigma$) Abs and MAC values of BrC at 365 nm (i.e., $Abs_{365,WS\text{-}BrC}$,
$Abs_{365,WI\text{-}BrC}$, $MAC_{365,WS\text{-}BrC}$, and $MAC_{365,WI\text{-}BrC}$) during daytime and nighttime in winter and
summer are summarized in Table 1. 365 nm is chosen to avoid interferences from inorganic
compounds (e.g., nitrate and nitrite) and to be consistence with previous studies (Hecobian et al.,
2010;Huang et al., 2018). On average, $Abs_{365,WS\text{-}BrC}$ is significantly higher than $Abs_{365,WI\text{-}BrC}$ in
summer ($5.00\pm1.28$ $Mm^{-1}$ vs. $2.95\pm1.94$ $Mm^{-1}$) but the values are comparable in winter ($19.6\pm8.3$
$Mm^{-1}$ vs. $21.9\pm13.5$ $Mm^{-1}$). The substantially higher BrC absorptions in winter correspond to a
much higher organic aerosol concentration – WSOC and WIOC concentrations in winter are on
average 4.2 and 14 times of the concentrations in summer (Table 1). Elevated OA (organic
aerosols) concentration during winter is due to a combination of lower ABL height and enhanced
primary emissions (e.g., from residential heating) in the cold season. It is worth noting that the
wavelength-dependent Abs of WS-BrC shows a minor tip at about 360 nm in both seasons
(Figure 2), which may be related to the contribution of some specific chromophores. For



example, Lin et al. (2015) reported that some nitrogen-containing organic compounds (such as
picric acid or nitrophenol) have a maximum absorption at wavelength of ~360 nm.
The MACs of WS-BrC are comparable between the two seasons (Figure 2c & d), with the
average $MAC_{365,WS-BrC}$ being 1.00 (±0.18) $m^2 g^{-1}$ in summer and 0.93 (±0.25) $m^2 g^{-1}$ in winter
(Table 1). As summarized in Table 2, the $MAC_{365,WS-BrC}$ measured in this study, i.e., at a rural
site in the Guanzhong Basin of China, is comparable to or lower than the values observed in
Asian cities such Xi'an (Huang et al., 2018), Beijing (Cheng et al., 2011), Seoul (Kim et al.,
2016) and New Delhi (Kirillova et al., 2014b). However, significantly lower $MAC_{365,WS-BrC}$
values were observed in the US, including Los Angeles Basin (Zhang et al., 2013), Southeastern
US (Hecobian et al., 2010), and Atlanta (Liu et al., 2013).
In winter, the average $MAC_{365,WI-BrC}$ (0.95±0.32 $m^2 g^{-1}$) is comparable to $MAC_{365,WS-BrC}$
(0.93±0.25 $m^2 g^{-1}$; Table 1). However, in summer the $MAC_{365,WI-BrC}$ is significantly higher than
$MAC_{365,WS-BrC}$ (1.82±1.06 vs. 1.00±0.18 $m^2 g^{-1}$), indicating a relatively stronger light absorption
capability of hydrophobic chromophores than hydrophilic chromophores. Further, the fact that
the summertime $MAC_{365,WI-BrC}$ is nearly double the wintertime $MAC_{365,WI-BrC}$ suggests that more
light absorbing molecules are formed in the warm season.
Figure 2 compares the wavelength-dependent light absorptivity (i.e., $Abs_\lambda$ and $MAC_\lambda$) of
WS-BrC and WI-BrC between day and night in summer and winter. Higher $Abs_{\lambda,WS-BrC}$ and
$Abs_{\lambda,WI-BrC}$ occurred during daytime in summer but during nighttime in winter. The $MAC_\lambda$ of
WS-BrC are overall similar between daytime and nighttime in both seasons. However, the $MAC_\lambda$
of WI-BrC show a significant daytime increase in summer over the whole wavelength range of
300-700 nm (Figure 2c). The day-night change of BrC light absorptivity can be viewed more



obviously in Figure 1e and 1f, where the temporal variations of the $Abs_{365}$ and $MAC_{365}$ of WS-
BrC and WI-BrC during summer 2016 (Aug. 3-23) and winter 2017 (Jan. 20 -Feb. 1) are
presented. The highest day/night ratio of $MAC_{365,WIOC}$ reached 3.8 in summer and the average
daytime $MAC_{365,WI\text{-}BrC}$ in summer ($2.45\pm1.14$ $m^2$ $g^{-1}$) is more than twice the value during
nighttime ($1.18\pm0.36$ $m^2$ $g^{-1}$; Table 1). A possible reason for this observation is that there are
additional sources of WI-BrC during summer daytime in this rural region, such as secondary
formation of hydrophobic light absorbing compounds.
Figure 3 and 4 present the cross-correlations of $Abs_{365,WS\text{-}BrC}$ and $Abs_{365,WI\text{-}BrC}$ with major
chemical components (e.g., WSOC, WSIC, and sulfate) and molecular tracer species in summer
and winter, respectively. In winter, $Abs_{365,WS\text{-}BrC}$ correlates strongly with WSOC concentration
($r^2=0.80$), so does $Abs_{365,WI\text{-}BrC}$ with WIOC ($r^2=0.76$). However, their relationships in summer are
much weaker, especially for the correlation between $Abs_{365,WI\text{-}BrC}$ and WIOC ($r^2=0.50$).
Considering that secondary OA (SOA) are mainly comprised of water-soluble compounds, such
as polyalcohols/polyacids and phenols (Kondo et al., 2007), the much higher WSOC/OC ratio in
summer ($0.75\pm0.07$) compared to winter ($0.50\pm0.09$) confirms more prevalent SOA formation in
summer associated with higher air temperature and stronger solar radiation. Formation of
secondary organic chromophores may lead to a more complex composition of BrC in summer.
More evidences on secondary BrC formation are provided in the subsequent sections.
Numerous studies reported that biomass burning is a dominant source of BrC in the
atmosphere (Desyaterik et al., 2013;Washenfelder et al., 2015). In the current study,
levoglucosan – a key tracer for biomass burning emissions (Simoneit, 2002) –was determined.
As shown in Figure 3 and 4, levoglucosan correlates well with WSOC and WIOC in both





summer and winter ($r^2$=0.45-0.77), suggesting that biomass burning is an important source of OA
in the rural region of Guanzhong Basin. For most of the periods in this study, the $MAC_{365,WS\text{-}BrC}$
and $MAC_{365,WI\text{-}BrC}$ values are within the range of MAC of biomass burning aerosols (e.g.,
1.3−1.8 for corn stalk (Li et al., 2016a), 1.37 for rice straw (Park and Yu, 2016), ~1.9 for BB
smoke particles (Lin et al., 2017)). Also, $Abs_{365,WI\text{-}BrC}$ in both summer and winter correlate well
with levoglusocan ($r^2$=0.74 and 0.62, respectively), demonstrating an important contribution of
biomass burning to WI-BrC despite the fact that levoglucosan itself is water soluble. The
relationships between the $Abs_{365,WS\text{-}BrC}$ and levoglucosan are much weaker ($r^2$=0.40 and 0.45 in
summer and winter, respectively), suggesting more complex sources of WS-BrC in the region.
**3.2 Molecular characterization of BrC aerosols**
Five categories of molecular tracer compounds, i.e., parent-polycyclic aromatic
hydrocarbons (parent-PAHs), oxygenated-PAHs (OPAHs), nitrophenols, isoprene-derived
products ($SOA_i$), and α-/β-pinene-derived products ($SOA_p$), were determined by the GC-EIMS
technique to investigate the formation pathways of BrC in this study. Their average
concentrations as well as daytime and nighttime differences are summarized in Table 1, and the
temporal variation profiles of the sum concentrations of each category, together with
levoglucosan time series, are presented in Figure S3.
PAHs and their derived compounds are important BrC chromophores, since the large
conjugated polycyclic structures are strongly light-absorbing in the near-UV range (Samburova
et al., 2016;Huang et al., 2018). A total number of 14 parent PAHs and 5 OPAHs (Table S1)
were determined in this study. Parent-PAHs are unsubstituted PAHs mainly emitted directly
from incomplete combustions of coal, biofuel, gasoline or other materials whereas OPAHs can



be emitted directly from combustion sources or formed from photochemical oxidation of the
parent-PAHs. The time trends of parent-PAHs and OPAHs are highly similar in both seasons ($r^2$
= 0.90 and 0.98 in summer and winter, respectively, Figure 3 and 4), suggesting that they have
common combustion sources. In addition, both parent-PAHs and OPAHs presented good
correlations with levoglucosan, particularly in winter ($r^2$ = 0.69 and 0.73, respectively; Figure 4),
indicating that that biomass burning is an important contributor to air particulate PAHs in the
region. PAHs, as well as levoglucosan, are elevated during nighttime in winter, corresponding to
enhanced biomass burning emissions from heating-related activities as well as reduced boundary
layer height at night. In contrast, the average daytime concentrations of parent-PAHs (11.6±5.7
ng m$^{-3}$) and levoglucosan (142±89 ng m$^{-3}$) in summer are about 1.95 and 2.58 times,
respectively, of the values at night (Table 1). The daytime enhancement of OPAHs
concentrations in summer is even more pronounced with an average day/night ratio of ~4.6 and
as high as 9.8 for individual OPAH species (e.g., 6H-henzo(cd)pyrene-6-one; Figure S4). Both
parent-PAHs and OPAHs, which are hydrophobic thus mainly exist as WIOC, demonstrate a
good linear relationship with $Abs_{365,WI-BrC}$ in both winter and summer ($r^2$ = 0.49-0.83, Figure 3
and 3). However, the good correlation between OPAHs and $Abs_{365,WI-BrC}$ in summer appears to
be mainly driven by daytime production, as the correlation coefficient ($r^2$) is 0.72 for the daytime
data but is <0.1 for the nighttime data (Figure S5a). These results suggest that photochemical
formation of light-absorption compounds is an important source of BrC during summer in the
Guanzhong Basin.

We estimated the potential contribution of parent-PAHs and OPAHs to the light absorption

of WI-BrC using a method reported in Samburova et al. (2016). Details on the method are



presented in the Supplementary Information (SI). Table S2 summarizes the solar-spectrum-
weighed mass absorption coefficients for PAHs ($MAC_{PAH,av}$) used in the calculation. As shown
in Figure 5, the contribution of parent-PAHs to solar-spectrum-weighed absorption coefficient of
WI-BrC varies between 0.55% - 0.66% with slight diurnal or season variations (Table S2).
However, the contribution of OPAHs clearly shows higher daytime values, especially in
summer. The average contribution of OPAHs to the solar-spectrum-weighed absorption
coefficient of WI-BrC in summer is 0.51±0.28% during daytime and 0.34±0.19% during
nighttime. These results indicate that more secondary water-insoluble aromatic chromophores
were produced via photochemical oxidation during summertime in the rural region.

Nitrophenols were identified as one of the most important light-absorbing compounds in

particles and cloud water influenced by BB emission in China (Desyaterik et al., 2013). These
compounds can be either directly emitted from burning of biomass (Xie et al., 2019) or formed in
the atmosphere through gas phase and aqueous phase reactions of aromatic precursors including
benz[a]pyrene (Lu et al., 2011), naphthalene (Kitanovski et al., 2014), catechol and guaiacol
(Ofner et al., 2011), and toluene (Liu et al., 2015) in the presence of $NO_x$. In this study, only a
few nitrophenol compounds were detected in PM (Table S1) and their average (±1σ)
concentration is 0.94 (±0.26) ng m$^{-3}$ in summer and 72.6 (±63.7) ng m$^{-3}$ in winter. The
wintertime concentrations of nitrophenols measured in the current study are comparable to those
detected in Shanghai (Li et al., 2016b), Mt. Tai in the Shandong province of China (Desyaterik et
al., 2013), and Ljubljana of Slovenia (Kitanovski et al., 2012), but the summertime
concentrations observed are more comparable to those detected in the Los Angeles Basin of the
U.S. (Zhang et al., 2013). The substantially lower concentration of nitrophenols in summer may



be related to rapid photodegradation in the atmosphere. Indeed, according to a laboratory study
conducted by Zhao et al. (2015) the timescale for photo-bleaching of nitrophenols can be an hour
or less. Furthermore, as shown in Figure S5b, during wintertime, when low temperature and
weak solar irradiation suppress photodegradation process, nitrophenols concentration anti-
correlates with $O_3$ mixing ratio in a nonlinear manner ($r^2$=0.60). On average, nitrophenols in
winter present 2.5 times higher concentration during nighttime than during daytime whereas the
nighttime concentrations of levoglucosan and PAHs are only slightly higher than the daytime
concentrations (by 11% and 33%, respectively; Table 1). Levoglucosan and PAHs are less
photochemically reactive than nitrophenols. These results confirm that nitrophenols, and other
photoreactive BrC compounds, may undergo significant atmospheric degradation during
summertime.

Both summertime and wintertime $Abs_{365,WS-BrC}$ correlate well with the concentrations of

nitrophenols ($r^2$=0.51-0.72, Figure S5c & d), suggesting an important contribution of nitrated
aromatic compounds to light absorption of WS-BrC in the study area. Using the MAC of
individual nitrophenol reported in Zhang et al. (2013), we calculated that the contributions of
nitrophenols to aerosol light absorption are 6.5-27 times higher than their mass contributions to
WSOC and that the fractions are much higher in winter (2.44±1.78%) than in summer
(0.12±0.03%; Table S3). In addition, due to a significantly higher abundance of nitrophenols
during nighttime in winter, their fractional contribution to aerosol absorption is on average 2.5
times higher than during the day (3.47±2.03% vs. 1.41±0.29%).

On a global scale, biogenic VOCs, mostly consisting of isoprene and monoterpenes, are

nearly an order of magnitude more abundant than anthropogenic VOCs (Guenther et al., 2006),



362 and their secondary products are estimated to be a predominant contributor to global SOA

363 burden (Heald et al., 2008). Recent studies (Lin et al., 2014;Nakayama et al., 2015;Nakayama et

364 al., 2012) showed that a large amount of biogenic SOA compounds are light absorptive. Some

365 tracers of SOA formed from isoprene ($SOA_i$) and α-/β-pinene ($SOA_p$) oxidation were measured

366 in the summertime samples (Table S1), and their temporal variations are shown in Figure S3. No

367 biogenic SOA tracer species were detectable in the winter samples in this study. Similar results

368 were obtained in our previous study in the Mt. Hua of the Guanzhong Basin (Li, 2011). These

369 findings are consistent with low emissions of biogenic VOCs and low oxidation rates in this

370 region during cold seasons. The average concentrations of $SOA_i$ and $SOA_p$ tracers in summer are

371 18.6±9.7 and 22.0±6.7 ng m$^{-3}$, respectively. Neither $SOA_i$ tracers nor $SOA_p$ tracers showed

372 significant correlations with the absorption coefficient of WSOC or WSIC, suggesting a low

373 contribution of biogenic SOA to aerosol light absorption in the region. In addition, compared to

374 the MAC values observed in this study, the MACs of biogenic SOA reported in literature are

375 much lower, on average by nearly an order of magnitude (Laskin et al., 2015), which further

376 support an insignificant contribution of biogenic sources to BrC in this region. This finding is

377 consistent with the fact that the Guanzhong Basin is a highly polluted region, where the major

378 emission sources of organic aerosols are anthropogenic.

379 **3.3 Variation of BrC during extreme haze events in winter**

380  In recent years, extreme haze events with very high $PM_{2.5}$ concentrations (up to 500-600 μg

381 m$^{-3}$) and low visibility (lower than 1 km) occurred frequently during wintertime in China (Huang

382 et al., 2014). In this study, an extreme haze event occurred during Jan. 21-26 when $PM_{2.5}$

383 concentration at the rural site increased continuously from ~100 μg m$^{-3}$ to 430 μg m$^{-3}$ and



visibility decreased from >10 km to ~1.4 km (Figure 1b & d). Similar to most haze events
occurred in Northeast China, this event was associated with stagnant meteorological condition
with low wind speed (<1 km s$^{-1}$) which promotes the accumulation of pollutants. In addition,
secondary inorganic aerosol species, e.g., $SO_4^{2-}$, $NO_3^-$ and $NH_4^+$, increased sharply (Figure 1d),
which indicates secondary aerosol formation was enhanced during the haze event despite the low
solar irradiance and low $O_3$ concertation (e.g., 2 ~ 40 μg m$^{-3}$; Figure 1c) conditions. Recent
studies by Wang et al. (2016) and Cheng et al. (2016) reported dramatic increases of secondary
inorganic components, mainly sulfate, nitrate and ammonium (SNA), during haze periods in
China and attributed the increases to enhanced aqueous reactions under high relative humidity
(RH) conditions with $NO_2$ being an important oxidant. Moreover, Huang et al. (2014) observed
that SOA also increased obviously during haze periods in winter. Indeed, as shown in Figure 4,
$SO_4^{2-}$ correlates well with RH (r$^2$=0.64) and $NO_2$ (r$^2$=0.56) in winter. In addition, $Abs_{365,WS-BrC,}$
which increases continuously during the haze period with a peak value at 43.3 Mm$^{-1}$ (Figure 1e),
correlates well with RH (r$^2$=0.65), sulfate (r$^2$=0.84) and $NO_2$ (r$^2$=0.70) (Figure 4). These results
suggest that aqueous oxidation has played a role in the formation of WS-BrC. This finding is
consistent with previous studies which have shown that aqueous reactions can be an important
pathway of BrC formation in the atmosphere (Laskin et al., 2015). In contrast, a slowly
decreasing trend of $MAC_{365,WIOC}$ is observed during the haze period, suggesting that some of the
water-insoluble BrC species were oxidized to form water-soluble chromophores, possibly
through aqueous-phase reactions.

It is worthwhile to mention that Jan. 27, 2017 was the Chinese New Year's Eve and a large

amount of fireworks were set off for celebration. During this night, the $MAC_{365,WS-BrC}$ (1.81)





increased to about 2 times of its average value in winter, while OC, EC, WSOC and WIOC as
well as SNA were actually 25%-51% lower than their wintertime average concentrations (Figure
1d). Meanwhile, metal ions which are abundant in fireworks (Wu et al., 2018;Jiang et al., 2015),
such as $K^+$, $Mg^{2+}$, $Ca^{2+}$, increased substantially during the night as well (Figure S6). These
results indicate that the increase of $MAC_{365,WSOC}$ during the Chinese New Year's Eve is likely
mainly contributed by metal-containing light-absorbing compounds emitted from fireworks
(Laskin et al., 2015;Tran et al., 2017).
**3.4 Estimation of direct absorption of solar radiation by BrC**

Since the light absorption of BrC is mainly in the UV spectral region, an important concern

is that BrC can reduce the solar actinic flux and thus affect atmospheric photochemistry and
tropospheric ozone production (Jacobson, 1998;Mohr et al., 2013). In this study, the direct
absorptions of solar radiation by both WS-BrC and WI-BrC are estimated using Eq 7. Figure S7
presents the incident solar irradiance and actinic flux spectra determined for the region under
midday summer (Aug. 10, 2016 13:00 pm Beijing (BJ) time) and winter (Jan. 25, 2017 13:00 pm
BJ time) conditions. Note that the local time at Guanzhong is ~ 1 hour later than the BJ time.

Table 3 presents a summary of the calculated direct solar absorptions of BrC. In summer,

the direct attenuation of actinic flux by WS-BrC and WI-BrC are estimated at
$1.55\times10^{14}\pm0.43\times10^{14}$ and $1.03\times10^{14}\pm0.64\times10^{14}$ quanta s$^{-1}$ cm$^{-2}$, respectively, in the UV range
(300-400 nm), which account for 0.83±0.23% and 0.53±0.33%, respectively, of the total down-
welling radiation. In winter, the direct absorptions by BrC are higher with WS-BrC and WI-BrC
on average account for 1.67±0.72% and 2.07±1.24%, respectively, of the total down-welling
radiation in the UV range. These results suggest that BrC may have a significant influence on



atmospheric photochemistry in the UV range. In the visible spectral region (400 - 700 nm), the
contributions of WS-BrC and WI-BrC to the total down-welling radiation are negligible –
0.10±0.03% and 0.07±0.05% in summer, and 0.15±0.06% and 0.15±0.08% in winter,
respectively.
Another concern of BrC is that they can absorb solar irradiance to influence tropospheric
temperature in a similar way as black carbon (BC) or elemental carbon (EC) (Feng et al.,
2013;Laskin et al., 2015). In our study, the direct absorption of solar irradiance by WS-BrC and
WI-BrC are estimated at 0.51±0.14 and 0.34±0.21 W m$^{-2}$ in summer, and 0.57±0.25 and
0.68±0.41 W m$^{-2}$ in winter in the UV range. To evaluate the contribution of BrC to total aerosol
absorption, we also estimated the direct absorption of EC based on the Carbon Analyzer data
according to the method described by Kirillova et al. (2014b) and Kirchstetter and Thatcher
(2012) (see SI). The estimated contributions of light absorption of BrC relative to EC are shown
in Table 3. In the visible region, the contribution is estimated at 10.0**±**3.52% in summer and
4.99±1.23% in winter for WS-BrC, and 6.19±2.42% and 4.51±1.44%, respectively, for WI-BrC.
However, in the UV range, the fractions increase to 49.3±14.5% in summer and 25.9±5.47% in
winter for WS-BrC, 29.4±11.0% and 29.0±10.4% for WI-BrC, which are within the range of the
values reported in other regions in China (Huang et al., 2018), India (Kirillova et al., 2014b), and
Korea (Kirillova et al., 2014a). On the other hand, the direct light absorption of WI-BrC
represents a substantive contribution to that of total BrC in this study, which is about 40% in
summer and more than 50% in winter in both UV and visible range, emphasizing the important
role that WI-BrC likely plays in atmospheric chemistry and the Earth's climate system,
especially in China.



## 4. Summary and Conclusion


Both WS-BrC and WI-BrC showed elevated Abs in winter (4-7 times higher than those in

summer), corresponding to much higher concentrations of WSOC and WIOC due to a
combination of lower ABL height and enhanced primary emissions (e.g., from residential
heating) in the cold season. No significant differences were found for the daytime and nighttime
MACs of WS-BrC in summer, or for the MACs of WS-BrC and WI-BrC in winter. However, the
average daytime $MAC_{365,WI\text{-}BrC}$ was more than twice the nighttime value in summer. We found
that the average daytime concentrations of both parent-PAHs and levoglucosan in summer were
around 2 times of the values at night and the daytime OPAHs concentration was more than 4
times of the nighttime value. Moreover, OPAHs and $Abs_{365,WI\text{-}BrC}$ correlated well during daytime
($r^2$=0.72) in summer but not during nighttime ($r^2$<0.1). These results demonstrated that
photochemical formation of BrC and enhanced BB emissions (e.g., from cooking) contributed to
the higher daytime MACs in summer. In winter, the Abs of WS-BrC correlated strongly with
relative humidity, sulfate, and $NO_2$, suggesting that aqueous-phase reactions played an important
role in the formation of secondary BrC. $Abs_{365,WS\text{-}BrC}$ correlated well with the concentrations of
nitrophenols in both seasons, suggesting an important contribution of nitrated aromatic
compounds to light absorption of WS-BrC. However, this contribution is much lower in summer
due to faster photodegradation reactions of these compounds. WS-BrC and WI-BrC were
estimated to account for 0.83±0.23% and 0.53±0.33%, respectively, of the total down-welling
solar radiation in the UV range in summer, and 1.67±0.72% and 2.07±1.24%, respectively, in
winter. The substantive contribution of WI-BrC to total BrC absorption (~40% in summer
and >50% in winter) emphasize the important role that WI-BrC likely plays in atmospheric



chemistry and the Earth's climate system.



**Author Contributions**
J.J. Li, Q. Zhang, G.H. Wang, K.F. Ho, and J.J. Cao designed the experiment. J.J. Li, G.H. Wang,
and K.F. Ho arranged the sample collection. J. Li, L. Liu and C. Wu collected the samples. J.J.
Li, J. Li, J.Y. Wang, W.Q. Jiang, and L.J. Li analyzed the samples. J.J. Li, Q. Zhang, and G.H.
Wang performed the data interpretation. J.J. Li, Q. Zhang, and G.H. Wang wrote the paper.

**Acknowledgements**
This work was financially supported by the program from National Nature Science Foundation
of China (No. 41773117, 91644102, 41977332, 91543116). The authors gratefully acknowledge
National Center for Atmospheric Research for the provision of the solar actinic flux and
irradiance data (TUV Quick Calculator,
http://cprm.acom.ucar.edu/Models/TUV/Interactive_TUV/) used in this publication.

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



Table 1 Average (±1σ) values Abs$_{365}$, MAC$_{365}$, and AAE of WS-BrC and WI-BrC, as well as concentrations of
OC, WSOC, WIOC, and measured organic species in the PM$_{2.5}$ aerosols from the rural site of Guanzhong Basin.

| | Summer | | | Winter | | |
|---|---|---|---|---|---|---|
| | Average | Daytime | Nighttime | Average | Daytime | Nighttime |
| Abs$_{365,WS-BrC}$ (Mm$^{-1}$) | 5.00±1.28 | 5.64±1.34 | 4.37±0.83 | 19.6±8.3 | 19.2±6.8 | 19.9±9.5 |
| Abs$_{365,WI-BrC}$ (Mm$^{-1}$) | 2.95±1.94 | 4.23±1.93 | 1.67±0.72 | 21.9±13.5 | 17.2±8.2 | 26.7±15.8 |
| MAC$_{365,WS-BrC}$ (m$^2$ g$^{-1}$) | 1.00±0.18 | 0.99±0.17 | 1.01±0.18 | 0.93±0.25 | 0.92±0.21 | 0.94±0.28 |
| MAC$_{365,WI-BrC}$ (m$^2$ g$^{-1}$) | 1.82±1.06 | 2.45±1.14 | 1.18±0.36 | 0.95±0.32 | 0.85±0.34 | 1.05±0.28 |
| AAE$_{WS-BrC}$ | 5.43±0.41 | 5.56±0.4 | 5.30±0.38 | 5.11±0.53 | 5.14±0.2 | 5.07±0.72 |
| AAE$_{WI-BrC}$ | 5.01±0.58 | 4.74±0.19 | 5.28±0.71 | 6.04±0.22 | 5.94±0.12 | 6.15±0.24 |
| OC (µg m$^{-3}$) | 6.78±1.77 | 7.74±1.73 | 5.83±1.19 | 45.9±22.9 | 44.0±17.2 | 47.9±27.2 |
| WSOC (µg m$^{-3}$) | 5.06±1.11 | 5.72±1.02 | 4.39±0.72 | 21.9±9.3 | 22.1±8.0 | 21.7±10.4 |
| WIOC (µg m$^{-3}$) | 1.73±0.87 | 2.02±1.04 | 1.44±0.53 | 24.0±14.3 | 21.9±10.1 | 26.2±17.3 |
| WSOC/OC | 0.75±0.07 | 0.75±0.09 | 0.76±0.04 | 0.50±0.09 | 0.51±0.08 | 0.48±0.10 |
| Parent-PAHs (ng m$^{-3}$) | 8.81±5.09 | 11.6±5.7 | 5.98±1.9 | 82.3±53.7 | 70.8±35.4 | 93.9±65.1 |
| OPAHs (ng m$^{-3}$) | 14.0±14.0 | 23.0±15.1 | 4.97±1.34 | 98.3±59.5 | 89.4±39.8 | 107±73 |
| Nitrophenols (ng m$^{-3}$) | 0.94±0.26 | 0.87±0.26 | 1.02±0.24 | 72.6±63.7 | 41.1±15.5 | 104±77 |
| SOA$_i$ [a] (ng m$^{-3}$) | 18.6±9.7 | 15.0±8.0 | 22.1±9.8 | BDL[c] | BDL | BDL |
| SOA$_p$ [b] (ng m$^{-3}$) | 22.0±6.7 | 25.2±6.7 | 18.9±5.0 | BDL | BDL | BDL |
| Levoglucosan (ng m$^{-3}$) | 98.7±83.7 | 142±89 | 55.1±48.7 | 601±301 | 569±138 | 633±401 |

[a] SOA$_i$: Tracers of SOA formed from isoprene (SOAi) oxidation, i.e., the sum of 2-methylglyceric acid, 2-methylthreitol, and 2-
methylerythritol.
[b] SOA$_i$: Tracers of SOA formed from α-/β-pinene (SOAp) oxidation, i.e., the sum of pinonic acid, pinic acid, and 3-methyl-1,2,3-
butanetricarboxylic acid.
[c] BDL: below detection limit (<0.17 ng m$^{-3}$) .


Table 2 Comparison of MAC$_{365,WS-BrC}$ in the present study and those reported in earlier studies in China, India, and
the United States (US).

| Sampling site | Sampling time | Season | MAC$_{365,WS-BrC}$ (m$^2$ g$^{-1}$) | Reference |
|---|---|---|---|---|
| Lincun, Shaanxi, China | Aug. 3-23, 2016 | Summer | 1.00±0.18 | This study |
| | Jan. 20-Feb 1, 2017 | Winter | 0.93±0.25 | |
| Xi'an, China | Jun. 1-Aug. 31, 2009 | Summer | 0.98 ±0.21 | Huang et al. (2018) |
| | Nov.15, 2008-Mar. 14, 2009 | Winter | 1.65 ± 0.36 | |
| Beijing, China | Jun. 20-Jul. 20, 2009 | Summer | 1.8 ± 0.2 | Cheng et al. (2011) |
| | Jan.9-Feb. 12, 2009 | Winter | 0.7 ± 0.2 | |
| Seoul, Korea | Aug. 13-Sep. 9, 2013 | Summer | 0.28 | Kim et al. (2016) |
| | Jan. 9-Feb 8, 2013 | Winter | 1.02 | |
| New Delhi, India | Oct. 24, 2010-Mar. 25, 2011 | Winter | 1.6 ± 0.5 | Kirillova et al. (2014b) |
| Los Angeles Basin, US | mid-May - mid-June, 2010 | summer | 0.71 | Zhang et al. (2013) |
| Southeastern US | 2007 | annually | 0.3–0.7 | Hecobian et al. (2010) |
| Atlanta, US | May 17-Sep. 29, 2012 | summer and fall | 0.14-0.53 | Liu et al. (2013) |







Table 3 Average direct solar absorption of water-soluble and water-insoluble BrC during summer and winter

| | WSOC | | WIOC | |
|---|---|---|---|---|
| | Summer | Winter | Summer | Winter |
| *Actinic flux ($\times 10^{14}$ quanta s$^{-1}$ cm$^{-2}$)* | | | | |
| 300-400 nm | 1.55±0.43 | 2.14±0.92 | 1.03±0.64 | 2.53±1.52 |
| 400-700 nm | 1.77±0.6 | 2.67±1.04 | 1.24±0.8 | 2.58±1.48 |
| *Irradiance (W m$^{-2}$)* | | | | |
| 300-400 nm | 0.51±0.14 | 0.57±0.25 | 0.34±0.21 | 0.68±0.41 |
| 400-700 nm | 0.49±0.17 | 0.57±0.22 | 0.35±0.23 | 0.55±0.32 |
| *Relative to EC (%)* | | | | |
| 300-400 nm | 49.4±14.5 | 25.9±5.47 | 29.4±11.0 | 29.0±10.4 |
| 400-700 nm | 10.0±3.52 | 4.99±1.23 | 6.19±2.42 | 4.51±1.44 |







**Figure Caption**
Figure 1 Temporal variation of meteorological parameters (a and b), concentrations of major
chemical compositions, $Abs_{365}$, $MAC_{365}$, and AAE of water-soluble and water-insoluble
BrC in $PM_{2.5}$ from the rural area of Northwest China.

Figure 2 Average spectra of absorption coefficient ($Abs_\lambda$) (a,b) and mass absorption coefficient
($MAC_\lambda$) (c,d) of water-soluble (WS-BrC) and water-insoluble (WI-BrC) BrC, as well as the
ratio of $MAC_{\lambda,WI\text{-}BrC}$ to $MAC_{\lambda,WI\text{-}BrC}$ (e,f) during daytime and nighttime of summer and
winter. Absorption Ångström exponent (AAE) is calculated by a linear regression of log
$Abs_\lambda$ versus log λ in the wavelength range of 300−450 nm.

Figure 3 Cross correlations between $Abs_{365,WS\text{-}BrC}$, $Abs_{365,WI\text{-}BrC}$, selected chemical compositions,
and RH in summer. The numbers at the upper right denote the linear correlation coefficients
($r^2$) of the corresponding scatter plots.

Figure 4 Cross correlations between $Abs_{365,WS\text{-}BrC}$, $Abs_{365,WI\text{-}BrC}$, selected chemical compositions,
and RH in winter. The numbers at the upper right denote the linear correlation coefficients
($r^2$) of the corresponding scatter plots.

Figure 5 Average contribution of parent-PAHs and OPAHs to the bulk light absorption of WI-
BrC (300−700 nm) during daytime and nighttime of summer and winter.






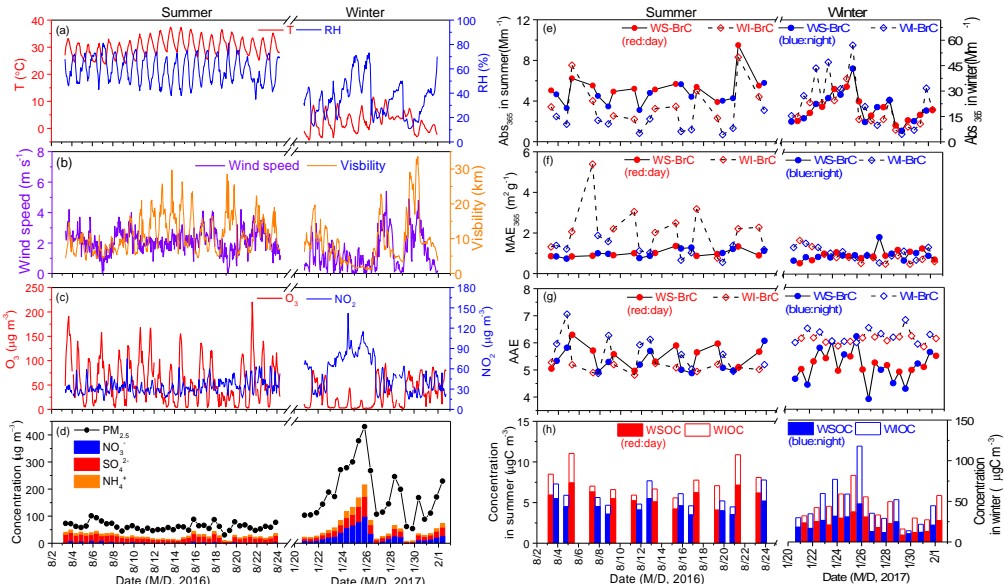


Figure 1 Temporal variation of meteorological parameters (a and b), concentrations of major chemical
compositions, Abs$_{365}$, MAC$_{365}$, and AAE of water-soluble and water-insoluble BrC in PM$_{2.5}$ from the rural
area of Northwest China.

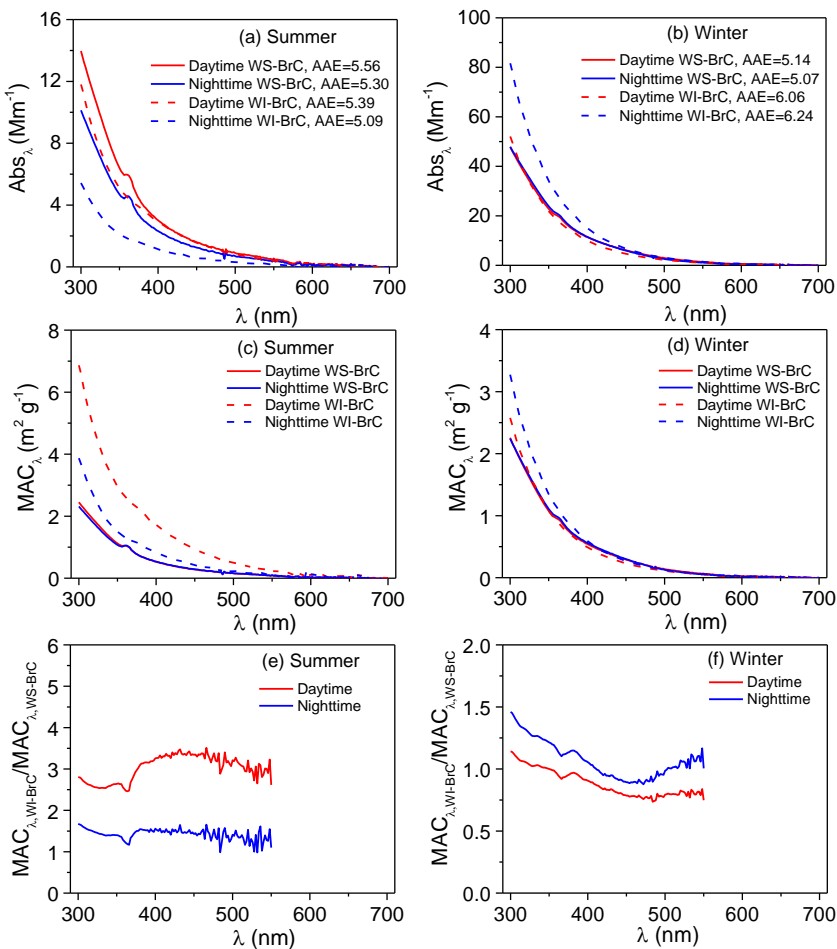


Figure 2 Average spectra of absorption coefficient (Abs$_\lambda$) (a,b) and mass absorption coefficient (MAC$_\lambda$) (c,d)

of water-soluble (WS-BrC) and water-insoluble (WI-BrC) BrC, as well as the ratio of MAC$_{\lambda,WI-BrC}$ to MAC$_{\lambda,WI-BrC}$ (e,f) during daytime and nighttime of summer and winter. Absorption Ångström exponent (AAE) is

calculated by a linear regression of log Abs$_\lambda$ versus log $\lambda$ in the wavelength range of 300−450 nm.


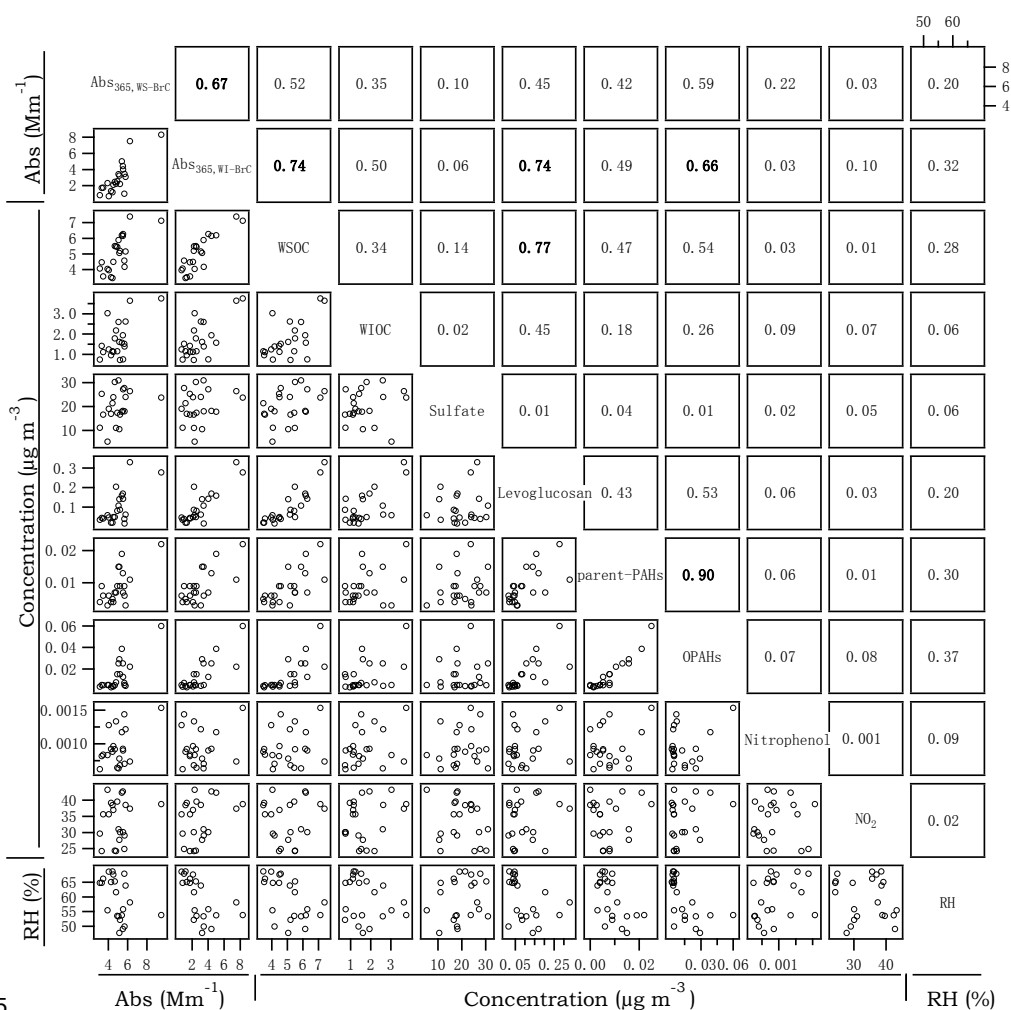

Figure 3 Cross-correlations between $Abs_{365,WS-BrC}$, $Abs_{365,WI-BrC}$, selected chemical components, and RH in
summer. The numbers at the upper right denote the linear correlation coefficients ($r^2$) of the corresponding
scatter plots.

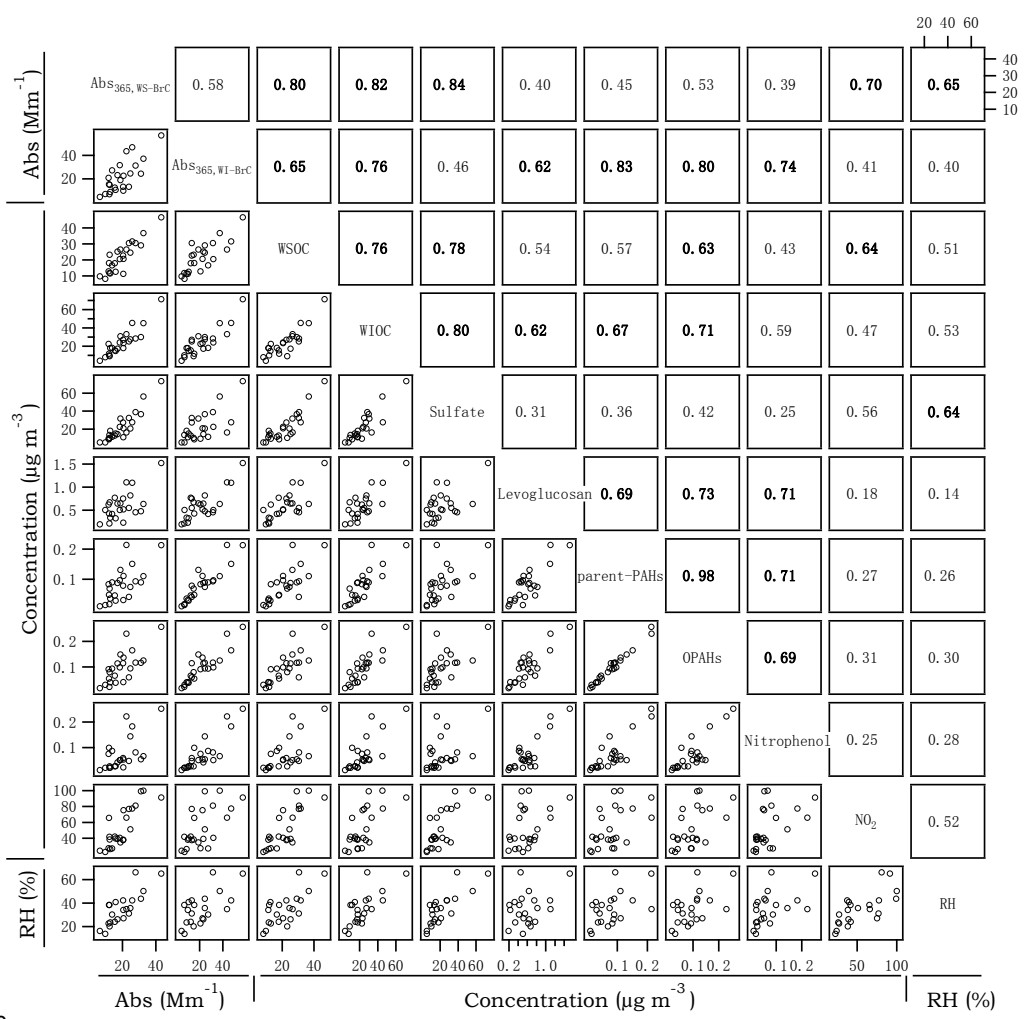


Figure 4 Cross-correlations between $Abs_{365,WS-BrC}$, $Abs_{365,WI-BrC}$, selected chemical components, and RH in

winter. The numbers at the upper right denote the linear correlation coefficients ($r^2$) of the corresponding

scatter plots.






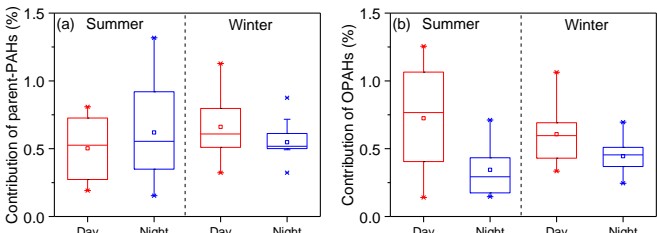


Figure 5 Average contribution of parent-PAHs and OPAHs to the bulk light absorption of WI-BrC (300−700
nm) during daytime and nighttime of summer and winter.
