# Peer review of "Optical properties and molecular compositions of water-soluble and water- insoluble brown carbon (BrC) aerosols in Northwest China"

_Atmospheric Chemistry and Physics, 2019_

## Referee Comment (RC1) · Anonymous Referee #3 · 31 Jan 2020

In this work, the authors examined the absorption properties and molecular compositions of water-soluble and –insoluble PM2.5 brown carbon from a rural site in China. Seasonal variation, day time vs night time, as well as water-soluble vs water-insoluble of absorbance and MAC values of particles were discussed. Their results showed the contribution of photochemical formation of Brown carbon and Biomass burning emissions to higher daytime MACs in summer in the region. They also suggest the important role of aqueous-phase reactions and nitrated aromatic compounds in the formation of secondary brown carbon. Overall, the authors have done a great job in analyzing and discussing their data. The work is also well presented. I recommend acceptance. Below are some minor comments.

[Figure]

1) Line 120 should it be "∼8am to 8pm"?

2) Section 2.4 please indicate where to subtract the signal from blanks in your calculations.

3) Line 170 what is M in Eq. 3?

4) Line 222 "Abs365 of WS-BrC is significantly higher than WI-BrC in summer, but values are comparable in winter". However, Figure 2 shows Abs365 of WI-BrC is higher than WS-BrC in winter. Please explain.

5) Line 405 an increase of MAC365 during New Year's Eve was observed but an increase of Abs365 or PM mass or WSOC was NOT observed. Please explain.

---

## Referee Comment (RC2) · Anonymous Referee #2 · 17 Feb 2020

This paper presents measurements of water soluble and insoluble brown carbon (BrC) in different seasons in Guangzhong Basin in China. The possible sources and radiative effects of BrC are also discusses. The paper is well written and within the scope of ACP. I would recommend the publication of this manuscript in ACP if the comments below are well addressed.

Specific comments:

L148: delete "and"

L171: Can the author give an estimate of the bias of $MAC_{\{\lambda,WI-BrC\}}$ in your measurement. It would be also good to provide measurement uncertainties of other derived

parameters.

L192: Can the author give some details of why 1.3 was used?

L203: how do these numbers compare with the measurement in other polluted regions (e.g. NCP, PRD and YRD) in China?

L219: Since there is a tip at 360 and definitely influences 365, maybe it is better to use another wavelength for reporting abs?

L221: I would not call it "significant higher"

L236: Since your site represents reginal background conditions (is it?), how these numbers compare with other reginal background measurements?

L311-315: Why is levoglucosan concentration much higher in daytime than in night-time in summer? Abs_{365,WI-BrC} shows higher R2 with levoglucosan concentration. Does it mean POA from biomass burning is also an important contributor to BrC in daytime in summer? I think based only on the correlations, it is difficult to judge if primary emission or photochemical formation is more important.

L398: Correlations with RH, sulfate and NO2 do not necessarily mean that aqueous oxidation has played a role in the formation of WS-BrC. We can see WI-BrC shows the same trend during the period. During haze event, the stagnant meteorological condition with low wind speed promotes the accumulation of BrC no matter how it is produced.

Figure 2e and f not mentioned in the text.

---

## Author Comment (AC1) · 19 Mar 2020

Dear Editor,

After reading the comments from the two referees, we have carefully revised our manuscript. Our response to the comments and related revisions are attached with this letter.

Anything about our paper, please feel free to contact me at dkwzhang@ucdavis.edu.

Best regards,

Sincerely yours
Qi Zhang Mar. 18, 2020

Please also note the supplement to this comment:
https://www.atmos-chem-phys-discuss.net/acp-2019-1002/acp-2019-1002-AC1-supplement.pdf
* * *
[Figure]

**Supplement:**

Journal: ACP
Title: Optical properties and molecular compositions of water-soluble and water-insoluble brown carbon (BrC) aerosols in Northwest China
Author(s): Jianjun Li et al.
MS No.: acp-2019-1002

Dear Editor,

After reading the comments from the two referees, we have carefully revised our manuscript. Our response to the comments and related revisions are attached with this letter.

Anything about our paper, please feel free to contact me at dkwzhang@ucdavis.edu.

Best regards,

Sincerely yours

Qi Zhang

Mar. 18, 2020

**Anonymous Referee #2**

This paper presents measurements of water soluble and insoluble brown carbon (BrC) in different seasons in Guangzhong Basin in China. The possible sources and radiative effects of BrC are also discusses. The paper is well written and within the scope of ACP. I would recommend the publication of this manuscript in ACP if the comments below are well addressed.

**Response:** We thank the referee's comments, which are very helpful for us to improve our work. Detailed revision and response to the comments are list below.

**Specific comments:**

L148: delete "and"

**Response:** Suggestion taken. Please see Line 150.

L171: Can the author give an estimate of the bias of $MAC_{\lambda, WI-BrC}$ in your measurement. It would be also good to provide measurement uncertainties of other derived parameters.

**Response:** We thank the referee's suggestion. We added the uncertainties of $Abs_{\lambda, WI-BrC}$ and $MAC_{\lambda, WI-BrC}$ in the revised manuscript. (Please see Line 173-174)

L192: Can the author give some details of why 1.3 was used?

**Response:** We thank the referee's suggestion. More details were provided in Line 196-205.

L203: how do these numbers compare with the measurement in other polluted regions (e.g. NCP, PRD and YRD) in China?

**Response:** We thank the referee's comment. We added a comparison of AAE value measured in this study with those in typical cities in NCP, PRD and YRD, please see Line 222-224.

L219: Since there is a tip at 360 and definitely influences 365, maybe it is better to use another wavelength for reporting abs?

**Response:** We do agree with the referee's comment. So we added an table (Table S1) to compare the Abs and MAC in wavelength of 340-380 nm. We found that the influence of the tip seems insignificant on the average $Abs_{365}$ or $MAC_{365}$. Thus, in order to be consistence with previous studies, we still use 365 nm for reporting Abs and MAC, and gave a detailed explanation at Line 251-254.

L221: I would not call it "significant higher"

**Response:** Suggestion taken. We revised to "much higher" in Line 266.

L236: Since your site represents reginal background conditions (is it?), how these numbers compare with other reginal background measurements?

**Response:** We thank the referee's comment. We added a comparison with some data reported in other regional or background sites in China, please see Line 260-262 and Table 2.

L311-315: Why is levoglucosan concentration much higher in daytime than in nighttime in summer? Abs_{365,WI-BrC} shows higher R2 with levoglucosan concentration. Does it mean POA from biomass burning is also an important contributor to BrC in daytime in summer? I think based only on the correlations, it is difficult to judge if primary emission or photochemical formation is more important.

**Response:** We thank the referee's comment. Yes, the concentration of levoglucosan is also higher in daytime than in nighttime in summer due to enhanced emission from BB for domestic cooking. As described in Line 303-305, "$Abs_{365,WI-BrC}$ in both summer and winter correlate well with levoglusocan ($r^2$=0.74 and 0.62, respectively), demonstrating an important contribution of biomass burning to WI-BrC…". However, we cannot quantitively judge if primary emission or photochemical formation is more important for WI-BrC in summer. Thus in "Conclusion", we concluded that "These results demonstrated that photochemical formation of BrC and enhanced BB emissions (e.g., from cooking) contributed to the higher daytime MACs in summer" (Line 495-497). Moreover, to avoid the confusion, we also revised the sentence "This difference was mainly attributed to enhanced photochemical formation of WI-BrC species, such as oxygenated polycyclic aromatic hydrocarbons (OPAHs)" in the Abstract to " This difference was partly attributed to enhanced photochemical formation of WI-BrC species, such as oxygenated polycyclic aromatic hydrocarbons (OPAHs)" (Line 42-43).

L398: Correlations with RH, sulfate and $NO_2$ do not necessarily mean that aqueous oxidation has played a role in the formation of WS-BrC. We can see WI-BrC shows the same trend during the period. During haze event, the stagnant meteorological condition with low wind speed promotes the accumulation of BrC no matter how it is produced.

**Response:** We thank the referee's comment. Indeed, the correlations of $Abs_{365,WS-BrC}$ with RH, sulfate and $NO_2$ are not necessarily to prove the aqueous formation of WS-BrC. However, we have more evidence for this. Firstly, the temporal variation of $Abs_{365,WI-BrC}$ and $Abs_{365,WS-BrC}$ is actually different. $Abs_{365,WS-BrC}$ increases continuously during the haze period, whereas $Abs_{365,WI-BrC}$ presents obvious diurnal variation (Figure 1). Secondly, the relationship of RH, sulfate and $NO_2$ with $Abs_{365,WS-BrC}$ are much stronger than those with $Abs_{365,WI-BrC}$, but levoglucosan and PAHs have stronger correlation with $Abs_{365,WI-BrC}$. These results suggested that $Abs_{365,WI-BrC}$ were more related with primary emissions, but aqueous oxidation was an important source for WS-BrC. In addition, in our previous study in Xi'an (~40 km away from the sampling site in this study), we further analyzed the stable carbon isotope composition and also found a secondary formation of BrC in winter (Wu et al., 2020). Thus, we confirmed that aqueous oxidation has played a role in the formation of WS-BrC. More detailed description also added in the manuscript, please see Line 423-430.

Figure 2e and f not mentioned in the text.
**Response:** We thank the referee's reminder. Figure 2e and f were deleted in the revised manuscript.

**Reference**

Wu, C., Wang, G., Li, J., Li, J., Cao, C., Ge, S., Xie, Y., Chen, J., Li, X., Xue, G., Wang, X., Zhao, Z., and Cao, F.: The characteristics of atmospheric brown carbon in Xi'an, inland China: sources, size distributions and optical properties, Atmos. Chem. Phys., 20, 2017-2030, 10.5194/acp-20-2017-2020, 2020.

**Anonymous Referee #3**

In this work, the authors examined the absorption properties and molecular compositions of water-soluble and –insoluble PM2.5 brown carbon from a rural site in China. Seasonal variation, day time vs night time, as well as water-soluble vs water-insoluble of absorbance and MAC values of particles were discussed. Their results showed the contribution of photochemical formation of brown carbon and biomass burning emissions to higher daytime MACs in summer in the region. They also suggest the important role of aqueous-phase reactions and nitrated aromatic compounds in the formation of secondary brown carbon. Overall, the authors have done a great job in analyzing and discussing their data. The work is also well presented. I recommend acceptance.

**Response:** We thank the referee's comments.

**Specific comments:**
1) Line 120 should it be "_8am to 8pm"?
**Response:** Suggestion taken. Please see Line 121.

2) Section 2.4 please indicate where to subtract the signal from blanks in your calculations.
**Response:** We thank the referee's comment. We added the information at Line 163-164.

3) Line 170 what is M in Eq. 3?
**Response:** We thank the referee's comment. We added an explanation about M in Eq. 3, please see 172.

4) Line 222 "$Abs_{365}$ of WS-BrC is significantly higher than WI-BrC in summer, but values are comparable in winter". However, Figure 2 shows $Abs_{365}$ of WI-BrC is higher than WS-BrC in winter. Please explain.
**Response:** We thank the referee's comment. The averaged value of $Abs_{365}$ of WI-BrC and WS-BrC in winter were $19.6\pm8.3$ $Mm^{-1}$ and $21.9\pm13.5$ $Mm^{-1}$, respectively. So we revised the sentence as "On average, $Abs_{365,WS\text{-}BrC}$ is significantly higher than $Abs_{365,WI\text{-}BrC}$ in summer ($5.00\pm1.28$ $Mm^{-1}$ vs. $2.95\pm1.94$ $Mm^{-1}$), but the values vary slightly in winter ($19.6\pm8.3$ $Mm^{-1}$ vs. $21.9\pm13.5$ $Mm^{-1}$)" (Line 240-242).

5) Line 405 an increase of $MAC_{365}$ during New Year's Eve was observed but an increase of $Abs_{365}$ or PM mass or WSOC was NOT observed. Please explain.
**Response:** We thank the referee's comment. The increase of $Abs_{365}$ or PM mass or WSOC was NOT observed because the metrological condition were favoring for pollutants dispersion (Figure 1). Thus, we provided more detailed explanation at Line 436-440.

[revised manuscript text omitted]
$_λ$ is the absorbance of the water (A$_{λ,WS\text{-}BrC}$) or ACN (A$_{λ,WI\text{-}BrC}$) extract at λ, which is corrected for the field blank,. V$_{solvent}$ (ml) is the volume of solvent (water or ACN) used to extract the filter (8 mL), and Va (m$^3$) is the air volume passed through the filter punch. $l$ (cm) is the optical length of the quartz cuvettes used for UV-vis measurement and ln(10) is used to convert the logbase-10 (provided by the spectrophotometer) to natural logarithm. 100 is for unit conversion. A$_{700}$ (absorbance at the wavelength of 700 nm) is subtracted to minimize the interference of baseline shift. The mass absorption coefficient of WS-BrC (MAC$_{\lambda,\text{WS-BrC}}$, m$^2$ g$^{-1}$)

or WI-BrC (MAC$_{\lambda,\text{WI-BrC}}$, m$^2$ g$^{-1}$) at wavelength of $\lambda$ is calculated using eq (3)

$$MAC_\lambda = \frac{Abs_\lambda}{M} \qquad (3)$$

where M is the mass concentration of WSOC or WIOC. Note that since it is possible that not all the WI-BrC was extracted into ACN, the Abs$_{\lambda,\text{WI-BrC}}$ (estimated uncertainty is 32%) and MAC$_{\lambda,\text{WI-}}$

$_\text{BrC}$ (estimated uncertainty is 33%) reported in this study are likely the lower bound values.

Nevertheless, the underestimation is probably insignificant since Chen and Bond (Chen and

Bond, 2010) reported that >92% of BrC was extractable by organic solvents (methanol or acetone).

The wavelength dependence for BrC absorption is fit with a power law equation:

$$Abs_\lambda = K \times \lambda^{-AAE} \qquad (4)$$

where K is a constant and AAE stands for absorption Ångström exponent. In this study, the AAE

for a given sample is calculated through the linear regression of log(Abs$_\lambda$) against log $\lambda$ between

300−450 nm. This wavelength range is chosen because the fits of all the samples in this study are better than $r^2$=0.99. Note that slightly higher AAE values (by up to 10%) are obtained using a wider wavelength range (e.g., 300-550 nm; Figure S2).

The fraction of solar irradiance absorbed by particulate BrC at a given wavelength $\lambda$ is estimated following the Beer−Lambert's law:

$$\frac{I_0 - I}{I_0}(\lambda) = 1 - e^{-b_{ap,\lambda,x} \times h_{ABL}} \qquad (5)$$

where $x$ denotes WS-BrC or WI-BrC, $h_{ABL}$ is the atmospheric boundary layer height (assuming

1200 m in summer and 600 m in winter) according to the assumption that the ground measurement results are representative of the average values in the whole planetary boundary layer (PBL)  (Kirchstetter et al.,

2004;Kirillova et al., 2014a). $I_0$ denotes the incident solar radiance in the form of either actinic flux (in quanta $s^{-1}$ $cm^{-2}$ $nm^{-1}$) or irradiance (in W $m^{-2}$ $nm^{-1}$), which were obtained using the TUV

Quick Calculator (http://cprm.acom.ucar.edu/Models/TUV/Interactive_TUV/). ($I_0$-I) denotes the direct absorption of solar actinic flux or irradiance by BrC. $b_{ap,\lambda,x}$ corresponds to the absorption coefficient ($b_{ap,}$ $m^{-1}$) of WS-BrC or WI-BrC at wavelength of $\lambda$. –The absorption properties of BrC extracted by bulk solution may not entirely reflect the light absorption by ambient aerosols. However, an estimated conversion factor can be calculated from the light absorption of size-resolved samples using the Mie theory. Assuming that particles are of spherical morphology and externally mixed with other light-absorbing components, an imaginary refractive index ($k$, responsible for absorption) could be obtained from MAC using follow equation (Laskin et al., 2015):

$$k_{(\lambda)} = \frac{\rho \times \lambda \times Abs_\lambda}{4\pi \times M_{WSOC}} = \frac{\rho \times \lambda \times MAC_\lambda}{4\pi} \qquad (6)$$

where $\rho$ (g/cm3) was particle density and assigned as 1.5, and more details about Mie theory calculations can be referred to the study by Liu et al. (2013). Previous studies showed that the light absorption coefficient of particulate BrC ($b_{ap,\lambda,BrC}$) is around 0.7−2.0 times of that from bulk solution ($Abs_{\lambda,WS-BrC \text{ or } WI-BrC}$) (Liu et al., 2013;Sun et al., 2007). Here, a conversion factor of 1.3

is applied based on a Mie theory calculation of aerosols in Xi'an (~ 40 km away from the sampling site) (Wu, 2018).

**3. Results and Discussion**

**3.1 Optical absorption characteristics of WS-BrC and WI-BrC**

The average absorption spectra of WS-BrC and WI-BrC ($\lambda$ = 300-700 nm) during daytime and nighttime in different seasons are shown in Figure 2a &b. The absorption Ångström exponents for both WS-BrC ($AAE_{WS-BrC}$) and WI-BrC ($AAE_{WI-BrC}$) are generally higher than 5, verifying the contribution of BrC to aerosol absorptivity in the region. The average $AAE_{WS-BrC}$

are similar between summer (5.43±0.41) and winter (5.11±0.53). Huang et al. (2014) and Shen et al. (2017) reported comparable $AAE_{WS-BrC}$ values (5.3-5.7) with no significant seasonal change at urban sites of Xi'an, suggesting common characteristics of BrC on a regional scale in the

Guanzhong Basin of China. These results are comparable with the data reported in Guangzhou (5.3) (Liu et al., 2018), but much lower than those in Beijing (5.3-7.3) (Cheng et al., 2011;Yan et al., 2015b;Du et al., 2014) and Nanjing (6.7-7.3) (Chen et al., 2018). Moreover, comparable AAE values were reported for WS-BrC in Switzerland (3.8-5.1) (Moschos et al.,

2018) and Nepal (4.2-5.6) (Wu et al., 2019;Kirillova et al., 2016), but higher $AAE_{WS-BrC}$ were found in  southeastern US (7 ± 1) (Hecobian et al., 2010), Los Angeles Basin (7.6 ±

0.5) (Zhang et al., 2013), and Korea (5.84-9.17) (Kim et al., 2016)

.

The $AAE_{WI-BrC}$ shows more obvious seasonal variations with a higher average value in winter (6.04±0.22) than in summer (5.01±0.58). This difference suggests that the chemical composition of WI-BrC might be more different in different seasons, due to variations in the sources and atmospheric formation and aging processes of light absorbing hydrophobic compounds.

The light absorption properties of WS-BrC and WI-BrC present obvious seasonal variations (Figure 2). The average (±1σ) Abs and MAC values of BrC at 365 nm (i.e., $Abs_{365,WS-BrC}$,

$Abs_{365,WI-BrC}$, $MAC_{365,WS-BrC}$, and $MAC_{365,WI-BrC}$) during daytime and nighttime in winter and summer are summarized in Table 1. 365 nm  was chosen to avoid interferences from inorganic compounds (e.g., nitrate and nitrite) and to be  consistent with previous studies (Hecobian et al., 2010;Huang et al., 2018). On average, $Abs_{365,WS-BrC}$ is significantly higher than

$Abs_{365,WI-BrC}$ in summer (5.00±1.28 $Mm^{-1}$ vs. 2.95±1.94 $Mm^{-1}$), but the values vary slightly in winter (19.6±8.3 $Mm^{-1}$ vs. 21.9±13.5 $Mm^{-1}$). The substantially higher

BrC absorptions in winter correspond to a much higher organic aerosol concentration – WSOC

and WIOC concentrations in winter are on average 4.2 and 14 times of the concentrations in summer (Table 1). Elevated OA (organic aerosols) concentration during winter is due to a combination of lower  PBL height and enhanced primary emissions (e.g., from residential heating) in the cold season. It is worth noting that the wavelength-dependent Abs of WS-BrC

shows a minor tip at about 360 nm in both seasons (Figure 2), which may be related to the contribution of some specific chromophores. For example, Lin et al. (2015) reported that some nitrogen-containing organic compounds (such as picric acid or nitrophenol) have a maximum absorption at wavelength of ~360 nm. The tip possibly caused an overestimation of average Abs and MAC at wavelength of 365 nm in this study. However, the influence seems insignificant based on a comparison of average Abs and MAC at wavelength of 340 nm, 350 nm,

360 nm, 370 nm, and 380 nm (Table S1).–

The MACs of WS-BrC are comparable between the two seasons (Figure 2c & d), with the average $MAC_{365,WS-BrC}$ being 1.00 ($\pm$0.18) $m^2$ $g^{-1}$ in summer and 0.93 ($\pm$0.25) $m^2$ $g^{-1}$ in winter (Table 1). As summarized in Table 2, the $MAC_{365,WS-BrC}$ measured in this study, i.e., at a rural site in the Guanzhong Basin of China, is comparable to or lower than the values observed in

Asian cities such Xi'an (Huang et al., 2018), Beijing (Cheng et al., 2011), Seoul (Kim et al.,

2016) and New Delhi (Kirillova et al., 2014b), but obviously higher than those in the regional sites of North China Plain (Teich et al., 2017) and the background site of Tibetan Plateau (Xu et al., 2020). HoweverMoreover, 
[revised manuscript text omitted]
 4). In contrast, $Abs_{365,WI-BrC}$ presents obvious diurnal variation during the haze period, and the  correlations of RH ($r^2$=0.40), sulfate ($r^2$=0.46) and $NO_2$

($r^2$=0.41) with $Abs_{365,WI-BrC}$ are also much weaker than those with $Abs_{365,WS-BrC}$. These results suggest that aqueous oxidation has played a role in the formation of WS-BrC (Laskin et al.,

2015) during the haze period, although the stagnant meteorological condition with low wind speed can also promote its accumulation. This finding is consistent with  our previous study conducted in Xi'an (Wu et al., 2020), which also found a secondary formation of

BrC in winter by using stable carbon isotope composition analysis

.

In contrast, a slowly decreasing trend of $MAC_{365,WIOC}$  was observed during the haze period, suggesting that some of the water-insoluble BrC species were oxidized to form water-soluble chromophores, possibly through aqueous-phase reactions.

It is worthwhile to mention that Jan. 27, 2017 was the Chinese New Year's Eve and a large amount of fireworks were set off for celebration. During this night, the concentrations of $PM_{2.5}$,

OC, EC, WSOC and WIOC as well as SNA were  25%-51% lower than their wintertime average concentrations due to the higher wind speed favoring for atmospheric dispersion (Figure

[revised manuscript text omitted]

De Haan, D. O., Tapavicza, E., Riva, M., Cui, T. Q., Surratt, J. D., Smith, A. C., Jordan, M. C., Nilakantan, S.,
Almodovar, M., Stewart, T. N., de Loera, A., De Haan, A. C., Cazaunau, M., Gratien, A., Pangui, E., and
Doussin, J. F.: Nitrogen-Containing, Light-Absorbing Oligomers Produced in Aerosol Particles Exposed to
Methylglyoxal, Photolysis, and Cloud Cycling, Environ. Sci. Technol., 52, 4061-4071,
10.1021/acs.est.7b06105, 2018.

Desyaterik, Y., Sun, Y., Shen, X., Lee, T., Wang, X., Wang, T., and Collett Jr., J. L.: Speciation of "brown" carbon in
cloud water impacted by agricultural biomass burning in eastern China, J. Geophys. Res.-Atmos., 118, 7389-
7399, doi:10.1002/jgrd.50561, 2013.

Du, Z., He, K., Cheng, Y., Duan, F., Ma, Y., Liu, J., Zhang, X., Zheng, M., and Weber, R.: A yearlong study of water-
soluble organic carbon in Beijing II: Light absorption properties, Atmos. Environ., 89, 235-241,
10.1016/j.atmosenv.2014.02.022, 2014.

[revised manuscript text omitted]

The characteristics of atmospheric brown carbon in Xi'an, inland China: sources, size distributions and optical properties, Atmos. Chem. Phys., 20, 2017-2030, 10.5194/acp-20-2017-2020, 2020.

Wu, G., Ram, K., Fu, P., Wang, W., Zhang, Y., Liu, X., Stone, E. A., Pradhan, B. B., Dangol, P. M., Panday, A. K.,

Wan, X., Bai, Z., Kang, S., Zhang, Q., and Cong, Z.: Water-Soluble Brown Carbon in Atmospheric Aerosols from Godavari (Nepal), a Regional Representative of South Asia, Environ. Sci. Technol., 53, 3471-3479,

10.1021/acs.est.9b00596, 2019.

Xie, M., Chen, X., Hays, M. D., and Holder, A. L.: Composition and light absorption of N-containing aromatic compounds in organic aerosols from laboratory biomass burning, Atmos. Chem. Phys., 19, 2899-2915,

10.5194/acp-19-2899-2019, 2019.

Xu, J., Cui, T. Q., Fowler, B., Fankhauser, A., Yang, K., Surratt, J. D., and McNeill, V. F.: Aerosol Brown Carbon from Dark Reactions of Syringol in Aqueous Aerosol Mimics, Acs Earth and Space Chemistry, 2, 608-617,

10.1021/acsearthspacechem.8b00010, 2018.

Xu, J., Hettiyadura, A. P. S., Liu, Y., Zhang, X., Kang, S., and Laskin, A.: Regional Differences of Chemical

Composition and Optical Properties of Aerosols in the Tibetan Plateau, Journal of Geophysical Research:

Atmospheres, 125, 10.1029/2019jd031226, 2020.

[revised manuscript text omitted]

---

## Author Response (AR1)

Journal: ACP Title: Optical properties and molecular compositions of water-soluble and waterinsoluble brown carbon (BrC) aerosols in Northwest China Author(s): Jianjun Li et al. MS No.: acp-2019-1002

Dear Editor,

After reading the comments from the two referees, we have carefully revised our manuscript. Our response to the comments and related revisions are attached with this letter.

Anything about our paper, please feel free to contact me at dkwzhang@ucdavis.edu.

Best regards,

Sincerely yours

Qi Zhang

Mar. 18, 2020

**Anonymous Referee #2**

This paper presents measurements of water soluble and insoluble brown carbon (BrC) in different seasons in Guangzhong Basin in China. The possible sources and radiative effects of BrC are also discusses. The paper is well written and within the scope of ACP. I would recommend the publication of this manuscript in ACP if the comments below are well addressed.

**Response:** We thank the referee's comments, which are very helpful for us to improve our work. Detailed revision and response to the comments are list below.

**Specific comments:**

L148: delete "and" **Response:** Suggestion taken. Please see Line 150.

L171: Can the author give an estimate of the bias of MAC\_{  $\lambda$ ,WI-BrC} in your measurement. It would be also good to provide measurement uncertainties of other derived parameters.

**Response:** We thank the referee's suggestion. We added the uncertainties of  $Abs_{\lambda,WI-BrC}$  and  $MAC_{\lambda,WI-BrC}$  in the revised manuscript. (Please see Line 173-174)

L192: Can the author give some details of why 1.3 was used? **Response:** We thank the referee's suggestion. More details were provided in Line 196-205.

L203: how do these numbers compare with the measurement in other polluted regions (e.g. NCP, PRD and YRD) in China?

**Response:** We thank the referee's comment. We added a comparison of AAE value measured in this study with those in typical cities in NCP, PRD and YRD, please see Line 222-224.

L219: Since there is a tip at 360 and definitely influences 365, maybe it is better to use another wavelength for reporting abs?

**Response:** We do agree with the referee's comment. So we added an table (Table S1) to compare the Abs and MAC in wavelength of 340-380 nm. We found that the influence of the tip seems insignificant on the average Abs365 or MAC365. Thus, in order to be consistence with previous studies, we still use 365 nm for reporting Abs and MAC, and gave a detailed explanation at Line 251-254.

L221: I would not call it "significant higher"

Response: Suggestion taken. We revised to "much higher" in Line 266.

L236: Since your site represents reginal background conditions (is it?), how these

numbers compare with other reginal background measurements? **Response:** We thank the referee's comment. We added a comparison with some data reported in other regional or background sites in China, please see Line 260-262 and Table 2.

L311-315: Why is levoglucosan concentration much higher in daytime than in nighttime in summer? Abs {365,WI-BrC} shows higher R2 with levoglucosan concentration. Does it mean POA from biomass burning is also an important contributor to BrC in daytime in summer? I think based only on the correlations, it is difficult to judge if primary emission or photochemical formation is more important. **Response:** We thank the referee's comment. Yes, the concentration of levoglucosan is also higher in daytime than in nighttime in summer due to enhanced emission from BB for domestic cooking. As described in Line 303-305, "Abs365,WI-BrC in both summer and winter correlate well with levoglusocan ( $r^2=0.74$  and 0.62, respectively), demonstrating an important contribution of biomass burning to WI-BrC...". However, we cannot quantitively judge if primary emission or photochemical formation is more important for WI-BrC in summer. Thus in "Conclusion", we concluded that "These results demonstrated that photochemical formation of BrC and enhanced BB emissions (e.g., from cooking) contributed to the higher daytime MACs in summer" (Line 495-497). Moreover, to avoid the confusion, we also revised the sentence "This difference was mainly attributed to enhanced photochemical formation of WI-BrC species, such as oxygenated polycyclic aromatic hydrocarbons (OPAHs)" in the Abstract to "This difference was partly attributed to enhanced photochemical formation of WI-BrC species, such as oxygenated polycyclic aromatic hydrocarbons (OPAHs)" (Line 42-43).

L398: Correlations with RH, sulfate and  $NO_2$  do not necessarily mean that aqueous oxidation has played a role in the formation of WS-BrC. We can see WI-BrC shows the same trend during the period. During haze event, the stagnant meteorological condition with low wind speed promotes the accumulation of BrC no matter how it is produced.

**Response:** We thank the referee's comment. Indeed, the correlations of Abs365,WS-BrC with RH, sulfate and NO2 are not necessarily to prove the aqueous formation of WS-BrC. However, we have more evidence for this. Firstly, the temporal variation of Abs365,WI-BrC and Abs365,WS-BrC is actually different. Abs365,WS-BrC increases continuously during the haze period, whereas Abs365,WI-BrC presents obvious diurnal variation (Figure 1). Secondly, the relationship of RH, sulfate and NO2 with Abs365,WS-BrC are much stronger than those with Abs365,WI-BrC, but levoglucosan and PAHs have stronger correlation with Abs365,WI-BrC. These results suggested that Abs365,WI-BrC were more related with primary emissions, but aqueous oxidation was an important source for WS-BrC. In addition, in our previous study in Xi'an (~40 km away from the sampling site in this study), we further analyzed the stable carbon isotope composition and also found a secondary formation of BrC in winter (Wu et al., 2020). Thus, we confirmed that aqueous oxidation has played a role in the

formation of WS-BrC. More detailed description also added in the manuscript, please see Line 423-430.

Figure 2e and f not mentioned in the text.

**Response:** We thank the referee's reminder. Figure 2e and f were deleted in the revised manuscript.

**Reference**

Wu, C., Wang, G., Li, J., Li, J., Cao, C., Ge, S., Xie, Y., Chen, J., Li, X., Xue, G., Wang, X., Zhao, Z., and Cao, F.: The characteristics of atmospheric brown carbon in Xi'an, inland China: sources, size distributions and optical properties, Atmos. Chem. Phys., 20, 2017-2030, 10.5194/acp-20-2017-2020, 2020.

**Anonymous Referee #3**

In this work, the authors examined the absorption properties and molecular compositions of water-soluble and –insoluble PM2.5 brown carbon from a rural site in China. Seasonal variation, day time vs night time, as well as water-soluble vs water-insoluble of absorbance and MAC values of particles were discussed. Their results showed the contribution of photochemical formation of brown carbon and biomass burning emissions to higher daytime MACs in summer in the region. They also suggest the important role of aqueous-phase reactions and nitrated aromatic compounds in the formation of secondary brown carbon. Overall, the authors have done a great job in analyzing and discussing their data. The work is also well presented. I recommend acceptance.

**Response:** We thank the referee's comments.

**Specific comments:**

1) Line 120 should it be "\_8am to 8pm"? **Response:** Suggestion taken. Please see Line 121.

2) Section 2.4 please indicate where to subtract the signal from blanks in your calculations.

**Response:** We thank the referee's comment. We added the information at Line 163-164.

3) Line 170 what is M in Eq. 3?

**Response:** We thank the referee's comment. We added an explanation about M in Eq. 3, please see 172.

4) Line 222 "Abs365 of WS-BrC is significantly higher than WI-BrC in summer, but values are comparable in winter". However, Figure 2 shows Abs365 of WI-BrC is higher than WS-BrC in winter. Please explain.

**Response:** We thank the referee's comment. The averaged value of  $Abs_{365}$  of WI-BrC and WS-BrC in winter were 19.6±8.3 Mm-1 and 21.9±13.5 Mm-1, respectively. So we revised the sentence as "On average,  $Abs_{365,WS-BrC}$  is significantly higher than  $Abs_{365,WI-BrC}$  in summer (5.00±1.28 Mm-1 vs. 2.95±1.94 Mm-1), but the values vary slightly in winter (19.6±8.3 Mm-1 vs. 21.9±13.5 Mm-1)" (Line 240-242).

5) Line 405 an increase of MAC365 during New Year's Eve was observed but an increase of Abs365 or PM mass or WSOC was NOT observed. Please explain. **Response:** We thank the referee's comment. The increase of Abs365 or PM mass or WSOC was NOT observed because the metrological condition were favoring for pollutants dispersion (Figure 1). Thus, we provided more detailed explanation at Line 436-440.

[revised manuscript text omitted]

15                                       |  <li>1 Key Lab of Aerosol Chemistry & Physics, SKLLQG, Institute of Earth Environment, Chinese Academy of Sciences, Xi'an 710061, China</li> <li>2 Department of Environmental Toxicology, University of California, Davis, CA 95616, USA</li> <li>3 Key Laboratory of Geographic Information Science of the Ministry of Education, School of Geographic Sciences, East China Normal University, Shanghai 200241, China</li> <li>4 Institute of Eco-Chongming, 3663 N. Zhongshan Rd., Shanghai 200062, China</li> <li>5 The Jockey Club School of Public Health and Primary Care, The Chinese University of Hong Kong, Hong Kong, China</li>  |
| 16                                                                               |                                                                                                                                                                                                                                                                                                                                                                                                                                                                                                                                                                                                                                                                                                                   |
| 17

30 | *Corresponding authors:
Prof. Qi Zhang
Department of Environmental Toxicology, University of California, Davis
One Shields Avenue, Davis, CA 95616
Phone: 1-530-752-5779
Fax: 1-530-752-3394
Email: dkwzhang@ucdavis.edu;
Prof. Gehui Wang
School of Geographic Sciences, East China Normal University, Shanghai, China
500 Dongchuan Rd., Shanghai 200241, China
Phone: 86-21-5434-1193
E-mail: ghwang@geo.ecnu.edu.cn.                                                                                                                                                                                                                                                         |

Brown carbon (BrC) contributes significantly to aerosol light absorption, thus can affect the earth's 32 33 radiation balance and atmospheric photochemical processes. In this study, we examined the light 34 absorption properties and molecular compositions of water-soluble (WS-BrC) and water-insoluble 35 (WI-BrC) BrC in PM2.5 collected from a rural site in the Guanzhong Basin – a highly polluted 36 region in Northwest China. Both WS-BrC and WI-BrC showed elevated light absorption 37 coefficients (Abs) in winter (4-7 times of those in summer) mainly attributed to enhanced emissions from residential biomass burning (BB) for house heating. While the average mass 38 39 absorption coefficients at 365 nm (MAC365) of WS-BrC were similar between daytime and nighttime in summer (0.99±0.17 and 1.01±0.18 m2 g-1, respectively), the average MAC365 of WI-40 BrC was more than a factor of 2 higher during daytime  $(2.45\pm1.14 \text{ m}^2 \text{ g}^{-1})$  than at night  $(1.18\pm0.36 \text{ m}^2)$ 41 m2 g-1). This difference was mainly partly 
[revised manuscript text omitted]

164 corrected for the field blank,  $V_{solvent}$  (ml) is the volume of solvent (water or ACN) used to

extract the filter (8 mL), and Va ( $m^3$ ) is the air volume passed through the filter punch. l (cm) is 165 the optical length of the quartz cuvettes used for UV-vis measurement and ln(10) is used to 166 convert the logbase-10 (provided by the spectrophotometer) to natural logarithm. 100 is for unit 167 conversion. A700 (absorbance at the wavelength of 700 nm) is subtracted to minimize the 168 interference of baseline shift. The mass absorption coefficient of WS-BrC (MAC $\lambda$ ,WS-BrC, m2 g-1) 169 or WI-BrC (MAC $\lambda$ ,WI-BrC, m2 g-1) at wavelength of  $\lambda$  is calculated using eq (3) 170  $MAC_{\lambda} = \frac{Abs_{\lambda}}{M}$ (3) 171 172 where M is the mass concentration of WSOC or WIOC. Note that since it is possible that not all the WI-BrC was extracted into ACN, the Abs $\lambda$ ,WI-BrC (estimated uncertainty is 32%) and MAC $\lambda$ ,WI- 173 174 Brc (estimated uncertainty is 33%) reported in this study are likely the lower bound values. Nevertheless, the underestimation is probably insignificant since Chen and Bond (Chen and 175 176 Bond, 2010) reported that >92% of BrC was extractable by organic solvents (methanol or 177 acetone). The wavelength dependence for BrC absorption is fit with a power law equation: 178  $Abs_{\lambda} = K \times \lambda^{-AAE}$ 179 (4) where K is a constant and AAE stands for absorption Ångström exponent. In this study, the AAE 180 for a given sample is calculated through the linear regression of  $log(Abs_{\lambda})$  against  $log \lambda$  between 181 300-450 nm. This wavelength range is chosen because the fits of all the samples in this study are 182 better than  $r^2=0.99$ . Note that slightly higher AAE values (by up to 10%) are obtained using a 183 wider wavelength range (e.g., 300-550 nm; Figure S2). 184 185 The fraction of solar irradiance absorbed by particulate BrC at a given wavelength  $\lambda$  is estimated following the Beer-Lambert's law: 186

187
$$\frac{I_0 - I}{I_0}(\lambda) = 1 - e^{-b_{ap,\lambda,x} \times h_{ABL}}$$
(5)

188 where *x* denotes WS-BrC or WI-BrC, hABL is the atmospheric boundary layer height (assuming
1200 m in summer and 600 m in winter) according to the assumption that the ground
190 measurement results are representative of the average values in the whole planetaryatmospheric
191 boundary layer (ABLPEL) (Kirchstetter et al., 2004;Kirillova et al., 2014a), (Kirchstetter et al.,
192 2004;Kirillova et al., 2014a). In denotes the incident solar radiance in the form of either actinic.
193 flux (in quanta s-1 cm-2 nm-1) or irradiance (in W m-2 nm-1), which were obtained using the TUV.
194 Quick Calculator (http://cprm.acom.ucar.edu/Models/TUV/Interactive\_TUV/). (Ig-1) denotes the
195 direct absorption of solar actinic flux or irradiance by BrC.end bap, x corresponds to the
196 absorption coefficient (bap, m-1) of WS-BrC or WI-BrC at wavelength of
$$\lambda_{--}$$
. The absorption
197 properties of BrC extracted by bulk solution may not entirely reflect the light absorption by
198 ambient aerosols. However, an estimated conversion factor can be calculated from the light.
199 absorption of size-resolved samples using the Mie theory. Assuming that particles are of.
200 spherical morphology and externally mixed with other light-absorbing components, an imaginary
201 refractive index (k, responsible for absorption) could be obtained from MAC using follow.
202 equation (Laskin et al., 2015):
203  $k_{(2)} = \frac{\rho \times \lambda \times Abs_{\lambda}}{4\pi \times M_{WSOC}} = \frac{\rho \times \lambda \times MAC_{\lambda}}{4\pi}$  (6)
204 where  $\rho$  (g/cm3) was particle density and assigned as 1.5, and more details about Mie theory
205 calculations can be referred to the study by Liu et al. (2013). Previous studies showed that the light
206 absorption coefficient of particulate BrC (bap, h, c) is around 0.7–2.0 times of that from bulk
207 solution (Absk, ws.Bac or WFBC ) (Liu et al., 2013;Sun et al., 2007). Here, a conversion factor of 1.3
208 is applied based on a Mie theory calculation of acrosols in Xi'an (~40 km away from the sampling

- 209 site) (Wu, 2018). He denotes the incident solar radiance in the form of either actinic flux (in quanta
- 210 s-1-cm-2-nm-1) or irradiance (in W m-2-nm-1), which were obtained using the TUV Quick Calculator
- 211 (http://cprm.acom.ucar.edu/Models/TUV/Interactive\_TUV/). (I0-I) denotes the direct absorption
- 212 of solar actinic flux or irradiance by BrC.
- 213 **3. Results and Discussion**

- 214 **3.1 Optical absorption characteristics of WS-BrC and WI-BrC**
- The average absorption spectra of WS-BrC and WI-BrC ( $\lambda = 300-700$  nm) during daytime 215 and nighttime in different seasons are shown in Figure 2a &b. The absorption Ångström 216 217 exponents for both WS-BrC (AAEWS-BrC) and WI-BrC (AAEWI-BrC) are generally higher than 5, verifying the contribution of BrC to aerosol absorptivity in the region. The average AAEWS-BrC 218 219 are similar between summer  $(5.43\pm0.41)$  and winter  $(5.11\pm0.53)$ . Huang et al. (2014) and Shen et 220 al. (2017) reported comparable AAEWS-BrC values (5.3-5.7) with no significant seasonal change at urban sites of Xi'an, suggesting common characteristics of BrC on a regional scale in the 221 222 Guanzhong Basin of China. These results are comparable with the data reported in Guangzhou 223 (5.3) (Liu et al., 2018), but much lower than those in Beijing (5.3-7.3) (Cheng et al., 2011; Yan et 224 al., 2015b;Du et al., 2014) and Nanjing (6.7-7.3) (Chen et al., 2018). Moreover, Comparable-225 comparable AAE values were reported for WS-BrC in Switzerland (3.8-5.1) (Moschos et al., 226 2018) and Nepal (4.2-5.6) (Wu et al., 2019;Kirillova et al., 2016), but higher AAEWS-BrC were 227 found in Southeastern southeastern US  $(7 \pm 1)$  (Hecobian et al., 2010), Los Angeles Basin  $(7.6 \pm 1)$ 0.5) (Zhang et al., 2013), and Korea (5.84-9.17) (Kim et al., 2016), and Beijing (7.0-7.5) (Cheng-228 229 et al., 2011).
  - The AAEWI-BrC shows more obvious seasonal variations with a higher average value in

231 winter  $(6.04\pm0.22)$  than in summer  $(5.01\pm0.58)$ . This difference suggests that the chemical composition of WI-BrC might be more different in different seasons, due to variations in the 232 233 sources and atmospheric formation and aging processes of light absorbing hydrophobic 234 compounds. 235 The light absorption properties of WS-BrC and WI-BrC present obvious seasonal variations (Figure 2). The average  $(\pm 1\sigma)$  Abs and MAC values of BrC at 365 nm (i.e., Abs365,WS-BrC, 236 Abs365,WI-BrC, MAC365,WS-BrC, and MAC365,WI-BrC) during daytime and nighttime in winter and 237 238 summer are summarized in Table 1. 365 nm is was chosen to avoid interferences from inorganic 239 compounds (e.g., nitrate and nitrite) and to be consistence consistent with previous studies 240 (Hecobian et al., 2010; Huang et al., 2018). On average, Abs365,WS-BrC is significantly higher than 241 Abs365,WI-BrC in summer  $(5.00\pm1.28 \text{ Mm}^{-1} \text{ vs. } 2.95\pm1.94 \text{ Mm}^{-1})$ , but the values arecomparable vary slightly in winter (19.6±8.3 Mm-1 vs. 21.9±13.5 Mm-1). The substantially higher 242 BrC absorptions in winter correspond to a much higher organic aerosol concentration – WSOC 243 and WIOC concentrations in winter are on average 4.2 and 14 times of the concentrations in 244 245 summer (Table 1). Elevated OA (organic aerosols) concentration during winter is due to a 246 combination of lower ABL PBL height and enhanced primary emissions (e.g., from residential 247 heating) in the cold season. It is worth noting that the wavelength-dependent Abs of WS-BrC shows a minor tip at about 360 nm in both seasons (Figure 2), which may be related to the 248 249 contribution of some specific chromophores. For example, Lin et al. (2015) reported that some nitrogen-containing organic compounds (such as picric acid or nitrophenol) have a maximum 250 251 absorption at wavelength of ~360 nm. The tip wouldpossibly caused an overestimation of 252 average Abs and MAC at wavelength of 365 nm in this study. However, the influence seems

253 insignificant based on a comparison of average Abs and MAC at wavelength of 340 nm, 350 nm,
 254 360 nm, 370 nm, and 380 nm (Table S1).-

| 255 | The MACs of WS-BrC are comparable between the two seasons (Figure 2c & d), with the                                                                                 |
|-----|---------------------------------------------------------------------------------------------------------------------------------------------------------------------|
| 256 | average MAC 365,WS-BrC being 1.00 ( $\pm$ 0.18) m 2 g -1 in summer and 0.93 ( $\pm$ 0.25) m 2 g -1 in winter |
| 257 | (Table 1). As summarized in Table 2, the MAC 365,WS-BrC measured in this study, i.e., at a rural                                                         |
| 258 | site in the Guanzhong Basin of China, is comparable to or lower than the values observed in                                                                         |
| 259 | Asian cities such Xi'an (Huang et al., 2018), Beijing (Cheng et al., 2011), Seoul (Kim et al.,                                                                      |
| 260 | 2016) and New Delhi (Kirillova et al., 2014b), but obviously higher than those in the regional                                                                      |
| 261 | sites of North China Plain (Teich et al., 2017) and the background site of Tibetan Plateau (Xu et                                                                   |

[revised manuscript text omitted]
_2$  (r2=0.70) (Figure 4). In contrast, Abs365,WI-BrC presents obvious diurnal variation during the 423 haze period, and the relationship correlations of RH ( $r^2=0.40$ ), sulfate ( $r^2=0.46$ ) and NO2 424  $(r^2=0.41)$  with Abs365,WI-BrC are also much weaker than those with Abs365,WS-BrC. These results 425 suggest that aqueous oxidation has played a role in the formation of WS-BrC (Laskin et al., 426 427 2015) during the haze period, although the stagnant meteorological condition with low wind 428 speed can also promote its accumulation. This finding is consistent with with-our previous

429 studiesstudy conducted in Xi'an (Wu et al., 2020), which also found a secondary formation of

430 BrC in winter by using stable carbon isotope composition analysis have shown that aqueous

431 reactions can be an important pathway of BrC formation in the atmosphere (Laskin et al., 2015).

- In contrast, a slowly decreasing trend of MAC365,WIOC is was observed during the haze period,
- 433 suggesting that some of the water-insoluble BrC species were oxidized to form water-soluble

434 chromophores, possibly through aqueous-phase reactions.

It is worthwhile to mention that Jan. 27, 2017 was the Chinese New Year's Eve and a large

amount of fireworks were set off for celebration. During this night, the concentrations of  $PM_{2.5}$ ,

- 437 OC, EC, WSOC and WIOC as well as SNA were actually 25%-51% lower than their wintertime
- 438 average concentrations due to the higher wind speed favoring for atmospheric dispersion (Figure

[revised manuscript text omitted]

- 534 Chen, Y., Ge, X., Chen, H., Xie, X., Chen, Y., Wang, J., Ye, Z., Bao, M., Zhang, Y., and Chen, M.: Seasonal light
  535 absorption properties of water-soluble brown carbon in atmospheric fine particles in Nanjing, China, Atmos.
  536 Environ., 187, 230-240, 10.1016/j.atmosenv.2018.06.002, 2018.
- Cheng, Y., He, K. B., Zheng, M., Duan, F. K., Du, Z. Y., Ma, Y. L., Tan, J. H., Yang, F. M., Liu, J. M., Zhang, X. L.,
  Weber, R. J., Bergin, M. H., and Russell, A. G.: Mass absorption efficiency of elemental carbon and watersoluble organic carbon in Beijing, China, Atmos. Chem. Phys., 11, 11497-11510, 10.5194/acp-11-11497-2011,
  2011.
- 541 Cheng, Y., Zheng, G., Wei, C., Mu, Q., Zheng, B., Wang, Z., Gao, M., Zhang, Q., He, K., Carmichael, G., Pöschl,
  542 U., and Su, H.: Reactive nitrogen chemistry in aerosol water as a source of sulfate during haze events in China,
  543 Science Advances, 2, 10.1126/sciadv.1601530, 10.1126/sciadv.1601530, 2016.
- 544 De Haan, D. O., Tapavicza, E., Riva, M., Cui, T. Q., Surratt, J. D., Smith, A. C., Jordan, M. C., Nilakantan, S.,
  545 Almodovar, M., Stewart, T. N., de Loera, A., De Haan, A. C., Cazaunau, M., Gratien, A., Pangui, E., and
  546 Doussin, J. F.: Nitrogen-Containing, Light-Absorbing Oligomers Produced in Aerosol Particles Exposed to
  547 Methylglyoxal, Photolysis, and Cloud Cycling, Environ. Sci. Technol., 52, 4061-4071,
- 548 10.1021/acs.est.7b06105, 2018.
- 549 Desyaterik, Y., Sun, Y., Shen, X., Lee, T., Wang, X., Wang, T., and Collett Jr., J. L.: Speciation of "brown" carbon in
  550 cloud water impacted by agricultural biomass burning in eastern China, J. Geophys. Res.-Atmos., 118, 7389551 7399, doi:10.1002/jgrd.50561, 2013.
- 552 Du, Z., He, K., Cheng, Y., Duan, F., Ma, Y., Liu, J., Zhang, X., Zheng, M., and Weber, R.: A yearlong study of water553 soluble organic carbon in Beijing II: Light absorption properties, Atmos. Environ., 89, 235-241,
  554 10.1016/j.atmosenv.2014.02.022, 2014.

[revised manuscript text omitted]
  The characteristics of atmospheric brown carbon in Xi'an, inland China: sources, size distributions and optical
  properties, Atmos. Chem. Phys., 20, 2017-2030, 10.5194/acp-20-2017-2020, 2020.
- Wu, G., Ram, K., Fu, P., Wang, W., Zhang, Y., Liu, X., Stone, E. A., Pradhan, B. B., Dangol, P. M., Panday, A. K.,
  Wan, X., Bai, Z., Kang, S., Zhang, Q., and Cong, Z.: Water-Soluble Brown Carbon in Atmospheric Aerosols
  from Godavari (Nepal), a Regional Representative of South Asia, Environ. Sci. Technol., 53, 3471-3479,
  10.1021/acs.est.9b00596, 2019.
- Xie, M., Chen, X., Hays, M. D., and Holder, A. L.: Composition and light absorption of N-containing aromatic
  compounds in organic aerosols from laboratory biomass burning, Atmos. Chem. Phys., 19, 2899-2915,
  10.5194/acp-19-2899-2019, 2019.
- Xu, J., Cui, T. Q., Fowler, B., Fankhauser, A., Yang, K., Surratt, J. D., and McNeill, V. F.: Aerosol Brown Carbon
  from Dark Reactions of Syringol in Aqueous Aerosol Mimics, Acs Earth and Space Chemistry, 2, 608-617,
  10.1021/acsearthspacechem.8b00010, 2018.
- Xu, J., Hettiyadura, A. P. S., Liu, Y., Zhang, X., Kang, S., and Laskin, A.: Regional Differences of Chemical
  Composition and Optical Properties of Aerosols in the Tibetan Plateau, Journal of Geophysical Research:
  Atmospheres, 125, 10.1029/2019jd031226, 2020.
- Yan, C., Zheng, M., Sullivan, A. P., Bosch, C., Desyaterik, Y., Andersson, A., Li, X., Guo, X., Zhou, T., Gustafsson,
  Ö., and Collett, J. L.: Chemical characteristics and light-absorbing property of water-soluble organic carbon in
  Beijing: Biomass burning contributions, Atmos. Environ., 121, 4-12,
  https://doi.org/10.1016/j.atmosenv.2015.05.005, 2015a.
- Yan, C., Zheng, M., Sullivan, A. P., Bosch, C., Desyaterik, Y., Andersson, A., Li, X., Guo, X., Zhou, T., Gustafsson,
  O., and Collett, J. L., Jr.: Chemical characteristics and light-absorbing property of water-soluble organic carbon
  in Beijing: Biomass burning contributions, Atmos. Environ., 121, 4-12, 10.1016/j.atmosenv.2015.05.005,
  2015b.
- Yan, C. Q., Zheng, M., Bosch, C., Andersson, A., Desyaterik, Y., Sullivan, A. P., Collett, J. L., Zhao, B., Wang, S.
  X., He, K. B., and Gustafsson, O.: Important fossil source contribution to brown carbon in Beijing during
  winter, Scientific reports, 7, DOI: 10.1038/srep43182, 10.1038/srep43182, 2017.
- Yu, L., Smith, J., Laskin, A., Anastasio, C., Laskin, J., and Zhang, Q.: Chemical characterization of SOA formed
  from aqueous-phase reactions of phenols with the triplet excited state of carbonyl and hydroxyl radical, Atmos.
  Chem. Phys., 14, 13801-13816, 10.5194/acp-14-13801-2014, 2014.
- Zhang, X., Lin, Y.-H., Surratt, J. D., and Weber, R. J.: Sources, Composition and Absorption Ångström Exponent of
   Light-absorbing Organic Components in Aerosol Extracts from the Los Angeles Basin, Environ. Sci. Technol.,
   47, 3685-3693, 10.1021/es305047b, 2013.
- Zhao, R., Lee, A. K. Y., Huang, L., Li, X., Yang, F., and Abbatt, J. P. D.: Photochemical processing of aqueous
  atmospheric brown carbon, Atmos. Chem. Phys., 15, 6087-6100, 10.5194/acp-15-6087-2015, 2015.
- 775

776 Table 1 Average  $(\pm 1\sigma)$  values Abs365, MAC365, and AAE of WS-BrC and WI-BrC, as well as concentrations of 777 OC, WSOC, WIOC, and measured organic species in the PM2.5 aerosols from the rural site of Guanzhong Basin.

|                                                     |           | Summer          |                 |                  | Winter          |                 |
|-----------------------------------------------------|-----------|-----------------|-----------------|------------------|-----------------|-----------------|
|                                                     | Average   | Daytime         | Nighttime       | Average          | Daytime         | Nighttime       |
| Abs 365,WS-BrC (Mm -1 )       | 5.00±1.28 | 5.64±1.34       | 4.37±0.83       | 19.6±8.3         | 19.2±6.8        | 19.9±9.5        |
| Abs 365,WI-BrC ( $Mm^{-1}$ )             | 2.95±1.94 | 4.23±1.93       | 1.67±0.72       | 21.9±13.5        | 17.2±8.2        | 26.7±15.8       |
| $MAC_{365,WS-BrC} (m^2 g^{-1})$                     | 1.00±0.18 | 0.99±0.17       | $1.01\pm0.18$   | 0.93±0.25        | 0.92±0.21       | $0.94 \pm 0.28$ |
| $MAC_{365,WI-BrC} (m^2 g^{-1})$                     | 1.82±1.06 | $2.45 \pm 1.14$ | 1.18±0.36       | $0.95 \pm 0.32$  | $0.85 \pm 0.34$ | $1.05 \pm 0.28$ |
| AAE WS-BrC                               | 5.43±0.41 | $5.56\pm0.4$    | $5.30 \pm 0.38$ | 5.11±0.53        | 5.14±0.2        | $5.07 \pm 0.72$ |
| AAEwi-BrC                                           | 5.01±0.58 | 4.74±0.19       | $5.28 \pm 0.71$ | 6.04±0.22        | 5.94±0.12       | 6.15±0.24       |
| OC (µg m -3 )                            | 6.78±1.77 | 7.74±1.73       | 5.83±1.19       | 45.9±22.9        | 44.0±17.2       | 47.9±27.2       |
| WSOC (µg m -3 )                          | 5.06±1.11 | 5.72±1.02       | 4.39±0.72       | 21.9±9.3         | 22.1±8.0        | 21.7±10.4       |
| WIOC (µg m -3 )                          | 1.73±0.87 | $2.02{\pm}1.04$ | 1.44±0.53       | 24.0±14.3        | 21.9±10.1       | 26.2±17.3       |
| WSOC/OC                                             | 0.75±0.07 | $0.75 \pm 0.09$ | $0.76 \pm 0.04$ | $0.50 \pm 0.09$  | $0.51 \pm 0.08$ | $0.48 \pm 0.10$ |
| Parent-PAHs (ng m -3 )                   | 8.81±5.09 | 11.6±5.7        | $5.98 \pm 1.9$  | 82.3±53.7        | $70.8 \pm 35.4$ | 93.9±65.1       |
| OPAHs (ng m -3 )                         | 14.0±14.0 | 23.0±15.1       | 4.97±1.34       | 98.3±59.5        | 89.4±39.8       | 107±73          |
| Nitrophenols (ng m -3 )                  | 0.94±0.26 | $0.87 \pm 0.26$ | 1.02±0.24       | 72.6±63.7        | 41.1±15.5       | 104±77          |
| SOA i a (ng m -3 ) | 18.6±9.7  | 15.0±8.0        | 22.1±9.8        | BDL c | BDL             | BDL             |
| SOA p b (ng m -3 ) | 22.0±6.7  | 25.2±6.7        | 18.9±5.0        | BDL              | BDL             | BDL             |
| Levoglucosan (ng m -3 )                  | 98.7±83.7 | 142±89          | 55.1±48.7       | 601±301          | 569±138         | 633±401         |

778 a SOAi: Tracers of SOA formed from isoprene (SOAi) oxidation, i.e., the sum of 2-methylglyceric acid, 2-methylthreitol, and 2-

779 methylerythritol.

780  $^{b}$  SOAi: Tracers of SOA formed from  $\alpha$ -/ $\beta$ -pinene (SOAp) oxidation, i.e., the sum of pinonic acid, pinic acid, and 3-methyl-1,2,3-

781 butanetricarboxylic acid.

782  $$^{\circ}\,BDL$:$  below detection limit (<0.17 ng m^-3) .

\_\_\_\_

783

Table 2 Comparison of MAC365,WS-BrC in the present study and those reported in earlier studies in China, India, and
 the United States (US).

| Sampling site                | Sampling time                      | Season                        | $MAC_{365,WS-BrC} (m^2 g^{-1})$ | Reference                                                                        |
|------------------------------|------------------------------------|-------------------------------|---------------------------------|----------------------------------------------------------------------------------|
| Lingun Chaonyi China         | Aug. 3-23, 2016                    | Summer                        | 1.00±0.18                       | This study.                                                                      |
| Lincun, Snaanxi, China       | Jan. 20-Feb 1, 2017                | Winter                        | 0.93±0.25                       | This study                                                                       |
| Vi'on China                  | Jun. 1-Aug. 31, 2009               | Summer                        | $0.98 \pm 0.21$                 | $\mathbf{H}_{\mathbf{u},\mathbf{o},\mathbf{n},\mathbf{c}} \text{ at al } (2018)$ |
| Al an, China                 | Nov.15, 2008-Mar. 14, 2009         | Winter                        | $1.65\pm0.36$                   | Huang et al. (2018)                                                              |
| Daiiing China                | Jun. 20-Jul. 20, 2009              | Summer                        | $1.8\pm0.2$                     | Change at al. $(2011)$                                                           |
| beijing, China               | Jan.9-Feb. 12, 2009                | Winter                        | $0.7\pm0.2$                     | Cheng et al. (2011)                                                              |
| XiangHe, Hebei, China        | Jul. 9-14 and Jul. 21-Aug. 1, 2013 | Summer_                       | $\underline{0.38\pm0.52^a}$     | Teich at al. $(2017)$                                                            |
| Wangdu, Hebei, China         | Jun. 4-24, 2014                    | Summer_                       | $\underline{0.55\pm0.15^a}$     | Telch et al. (2017)                                                              |
| Mt. Waliguan, Qinghai, China | Jul. 1–31, 2017                    | Summer_                       | 0.48                     | Xu et al. (2020)                                                                 |
| Sacul Koran                  | Aug. 13-Sep. 9, 2013               | Summer                        | 0.28                            | Kim at al. $(2016)$                                                              |
| Scoul, Kolea                 | Jan. 9-Feb 8, 2013                 | Winter                        | 1.02                            | Killi et al. (2010)                                                              |
| New Delhi, India             | Oct. 24, 2010-Mar. 25, 2011        | Winter                        | $1.6\pm0.5$                     | Kirillova et al. (2014b)                                                         |
| Los Angeles Basin, US        | mid-May - mid-June, 2010           | summerSummer                  | 0.71                            | Zhang et al. (2013)                                                              |
| Southeastern US              | 2007                               | annually Annually      | 0.3–0.7                         | Hecobian et al. (2010)                                                           |
| Atlanta, US                  | May 17-Sep. 29, 2012               | summer-Summer
and fallFall | 0.14-0.53                       | Liu et al. (2013)                                                                |

a Data at XiangHe and Wangdu were the averaged MAC of WSOC at wavelength of 370 nm (i.e., MAC370,WS-BrC)
* * *
**792 Table 3 Average direct solar absorption of water-soluble and water-insoluble BrC during summer and winter**

|                                   | V                                         | VSOC      | WIOC               |                 |  |  |
|-----------------------------------|-------------------------------------------|-----------|--------------------|-----------------|--|--|
|                                   | Summer                                    | Winter    | Summer             | Winter          |  |  |
| Actinic flux ( $\times 10^{14}$ c | quanta s -1 cm -2 ) |           |                    |                 |  |  |
| 300-400 nm                        | 1.55±0.43                                 | 2.14±0.92 | 1.03±0.64          | 2.53±1.52       |  |  |
| 400-700 nm                        | 1.77 ± 0.6                         | 2.67±1.04 | 1.24±0.8           | 2.58±1.48       |  |  |
| Irradiance (W m -2 )   |                                           |           |                    |                 |  |  |
| 300-400 nm                        | 0.51±0.14                                 | 0.57±0.25 | 0.34±0.21          | $0.68 \pm 0.41$ |  |  |
| 400-700 nm                        | 0.49±0.17                                 | 0.57±0.22 | 0.35±0.23          | 0.55±0.32       |  |  |
| Relative to EC (%)                |                                           |           |                    |                 |  |  |
| 300-400 nm                        | 49.4±14.5                                 | 25.9±5.47 | 29.4±11.0          | 29.0±10.4       |  |  |
| 400-700 nm                        | $10.0 \pm 3.52$                           | 4.99±1.23 | 6.19 ± 2.42 | 4.51±1.44       |  |  |

| 795 | Figure Caption                                                                                                                                     |
|-----|----------------------------------------------------------------------------------------------------------------------------------------------------|
| 796 | Figure 1 Temporal variation of meteorological parameters (a and b), concentrations of major                                                        |
| 797 | chemical compositions, Abs 365 , MAC 365 , and AAE of water-soluble and water-insoluble                                      |
| 798 | BrC in PM 2.5 from the rural area of Northwest China.                                                                                   |
| 799 |                                                                                                                                                    |
| 800 | Figure 2 Average spectra of absorption coefficient (Abs $_{\lambda}$ ) (a,b) and mass absorption coefficient                                       |
| 801 | $(MAC_{\lambda})$ (c,d) of water-soluble (WS-BrC) and water-insoluble (WI-BrC) BrC, as well as the-                                                |
| 802 | ratio of MAC <math>\lambda</math>,WI-BrC to MAC <math>\lambda</math>,WI-BrC (e,f) during daytime and nighttime of summer and |
| 803 | winter. Absorption Ångström exponent (AAE) is calculated by a linear regression of log                                                             |
| 804 | Abs <math>\lambda</math> versus log $\lambda$ in the wavelength range of 300–450 nm.                                                    |
| 805 |                                                                                                                                                    |
| 806 | Figure 3 Cross correlations between Abs 365,WS-BrC , Abs 365,WI-BrC , selected chemical compositions,                        |
| 807 | and RH in summer. The numbers at the upper right denote the linear correlation coefficients                                                        |
| 808 | $(r^2)$ of the corresponding scatter plots.                                                                                                        |
| 809 |                                                                                                                                                    |
| 810 | Figure 4 Cross correlations between Abs 365,WS-BrC , Abs 365,WI-BrC , selected chemical compositions,                        |
| 811 | and RH in winter. The numbers at the upper right denote the linear correlation coefficients                                                        |
| 812 | $(r^2)$ of the corresponding scatter plots.                                                                                                        |
| 813 |                                                                                                                                                    |
| 814 | Figure 5 Average contribution of parent-PAHs and OPAHs to the bulk light absorption of WI-                                                         |
| 815 | BrC (300–700 nm) during davtime and nighttime of summer and winter.                                                                                |
| 040 |                                                                                                                                                    |
| 810 |                                                                                                                                                    |
| 817 |                                                                                                                                                    |
| 818 |                                                                                                                                                    |